# Arterial smooth muscle cell PKD2 (TRPP1) channels regulate systemic blood pressure

Simon Bulley[†], Carlos Fernández-Peña[†], Raquibul Hasan, M Dennis Leo, Padmapriya Muralidharan, Charles E Mackay, Kirk W Evanson, Luiz Moreira-Junior, Alejandro Mata-Daboin, Sarah K Burris, Qian Wang, Korah P Kuruvilla, Jonathan H Jaggar*

Department of Physiology, University of Tennessee Health Science Center, Memphis, United States

**Abstract** Systemic blood pressure is determined, in part, by arterial smooth muscle cells (myocytes). Several Transient Receptor Potential (TRP) channels are proposed to be expressed in arterial myocytes, but it is unclear if these proteins control physiological blood pressure and contribute to hypertension in vivo. We generated the first inducible, smooth muscle-specific knockout mice for a TRP channel, namely for PKD2 (TRPP1), to investigate arterial myocyte and blood pressure regulation by this protein. Using this model, we show that intravascular pressure and $\alpha_1$-adrenoceptors activate PKD2 channels in arterial myocytes of different systemic organs. PKD2 channel activation in arterial myocytes leads to an inward $Na^+$ current, membrane depolarization and vasoconstriction. Inducible, smooth muscle cell-specific PKD2 knockout lowers both physiological blood pressure and hypertension and prevents pathological arterial remodeling during hypertension. Thus, arterial myocyte PKD2 controls systemic blood pressure and targeting this TRP channel reduces high blood pressure.

*For correspondence:
jjaggar@uthsc.edu

[†]These authors contributed equally to this work

Competing interests: The authors declare that no competing interests exist.

## Introduction

Systemic blood pressure is controlled by total peripheral resistance, which is determined by the diameter of small arteries and arterioles. Arterial smooth muscle cell (myocyte) contraction reduces luminal diameter, leading to an increase in systemic blood pressure, whereas relaxation results in vasodilation that decreases blood pressure. Membrane potential is a primary determinant of arterial contractility (*Nelson et al., 1990*). Depolarization activates myocyte voltage-dependent calcium ($Ca^{2+}$) channels ($Ca_V$), leading to an increase in intracellular $Ca^{2+}$ concentration $[Ca^{2+}]_i$ and vasoconstriction. In contrast, hyperpolarization reduces $[Ca^{2+}]_i$, resulting in vasodilation (*Nelson et al., 1990*). In vitro studies have identified several different types of ion channel that regulate the membrane potential and contractility of arterial myocytes, but whether many of these ion channels regulate systemic blood pressure is unclear. Given that hypertension is associated with altered arterial contractility, myocyte ion channels that contribute to high blood pressure are also important to determine.

Transient Receptor Potential (TRP) channels constitute a family of ~28 proteins that are subdivided into six different classes, including polycystin (TRPP), canonical (TRPC), vanilloid (TRPV), ankyrin (TRPA), and melastatin (TRPM) (*Earley and Brayden, 2015*). Experiments performed using cultured and non-cultured cells and whole arteries, which contain multiple different cell types, have suggested that approximately thirteen different TRP channels may be expressed in arterial myocytes, including PKD2 (also termed TRPP1), TRPC1, TRPC3-6, TRPV1-4, TRPA1, TRPM4 and TRPM8 (*Earley and Brayden, 2015*). In many of these studies, TRP channel expression was reported in

myocytes of vasculature that does not control systemic blood pressure, including conduit vessels, cerebral arteries, portal vein and pulmonary arteries (*Earley and Brayden, 2015*). Blood pressure measurements in constitutive, global TRPC6, TRPM4 and TRPV4 channel knockout mice produced inconsistent results or generated complex findings that were associated with compensatory mechanisms (*Mathar et al., 2010*; *Dietrich et al., 2005*; *Nishijima et al., 2014*; *Earley et al., 2009*). Thus, the contribution of arterial myocyte TRP channels to physiological systemic blood pressure and pathological changes in blood pressure are unclear. Also uncertain is whether TRP channel subtypes present in arterial myocytes of different organs are similarly regulated by physiological stimuli and if such modulation leads to the same functional outcome.

PKD2 is a six transmembrane domain protein with cytoplasmic N and C termini (*Shen et al., 2016*). PKD2 is expressed in myocytes of rat and human cerebral arteries, mouse and human mesenteric arteries and in porcine whole aorta (*Griffin et al., 1997*; *Torres et al., 2001*; *Narayanan et al., 2013*). RNAi-mediated knockdown of PKD2 inhibited pressure-induced vasoconstriction (myogenic tone) in cerebral arteries (*Narayanan et al., 2013*; *Sharif-Naeini et al., 2009*). In global, constitutive PKD2$^{+/-}$ mice, an increase in actin and myosin expression lead to larger phenylephrine-induced contractions in aorta and mesenteric arteries (*Qian et al., 2007*; *Du et al., 2010*). It has also been proposed that arterial myocyte PKD2 channels inhibit myogenic tone in mesenteric arteries (*Sharif-Naeini et al., 2009*). Thus, in vitro studies have generated variable findings regarding physiological functions of arterial myocyte PKD2 channels.

Here, we generated an inducible, myocyte-specific PKD2 knockout mouse to investigate blood pressure regulation by this TRP channel. We show that inducible and cell-specific PKD2 channel knockout in myocytes reduces systemic blood pressure. We demonstrate that intravascular pressure and $\alpha_1$-adrenoceptor activation activate myocyte PKD2 channels in systemic arteries of different tissues, leading to an inward sodium current ($I_{Na}$), membrane depolarization and vasoconstriction. We also show that PKD2 channels are upregulated during hypertension and that myocyte PKD2 knockout causes vasodilation, attenuates remodeling of the arterial wall and reduces high blood pressure during hypertension. In summary, our study indicates that PKD2 channels in systemic artery myocytes control physiological blood pressure and are upregulated during hypertension, contributing to the blood pressure elevation.

## Results

### Generation of tamoxifen-inducible smooth muscle-specific PKD2 knockout mice

Mice with *loxP* sites flanking exons 11 and 13 (*Pkd2*$^{fl/fl}$) of the *Pkd2* gene were crossed with tamoxifen-inducible *Myh11-cre/ERT2* mice, producing a *Pkd2*$^{fl/fl}$:*Myh11-cre/ERT2* line. Genotyping was performed using a 3-primer (a, b, c) strategy to identify wild-type (in C57BL/6J), floxed and deleted PKD2 alleles. PCR of genomic DNA from mesenteric and hindlimb arteries of wild-type mice that lack *loxP* sites produced a 232 bp transcript (*Figure 1—figure supplement 1*). Tamoxifen-treated *Pkd2*$^{fl/fl}$ mouse arteries and aorta produced a transcript of 318 bp, which arose from primers a and b (*Figure 1—figure supplement 1*). PCR amplified a 209 bp transcript in vasculature from tamoxifen-treated *Pkd2*$^{fl/fl}$:*Myh11-cre/ERT2* mice due to primers a and c, confirming loss of the primer b annealing site. PCR of vasculature in *Pkd2*$^{fl/fl}$:*Myh11-cre/ERT2* arteries also produced a faint 318 bp band, suggesting that PKD2 is expressed in vascular wall cell types other than myocytes where DNA would not undergo recombination (*Figure 1—figure supplement 1*).

### PKD2 transcripts and protein are absent in arterial myocytes of tamoxifen-treated *Pkd2*$^{fl/fl}$:*Myh11-cre/ERT2* mice

RT-PCR was performed on RNA extracted from arterial myocytes (~500 cells) that had been individually harvested from mesenteric arteries of tamoxifen-treated *Pkd2*$^{fl/fl}$ and tamoxifen-treated *Pkd2*$^{fl/fl}$:*Myh11-cre/ERT2* mice. Arterial myocyte cDNA from both genotypes amplified transcripts for both actin and myosin heavy chain 11, a smooth muscle-specific marker (*Figure 1A*). In contrast, products for PECAM, an endothelial cell marker, and aquaporin 4, an astrocyte marker, were not amplified, suggesting that the isolated mRNA was from pure smooth muscle (*Figure 1A*). PKD2 primers were designed to anneal to a sequence in exons 9/10 (forward) and 13 (reverse), which span the

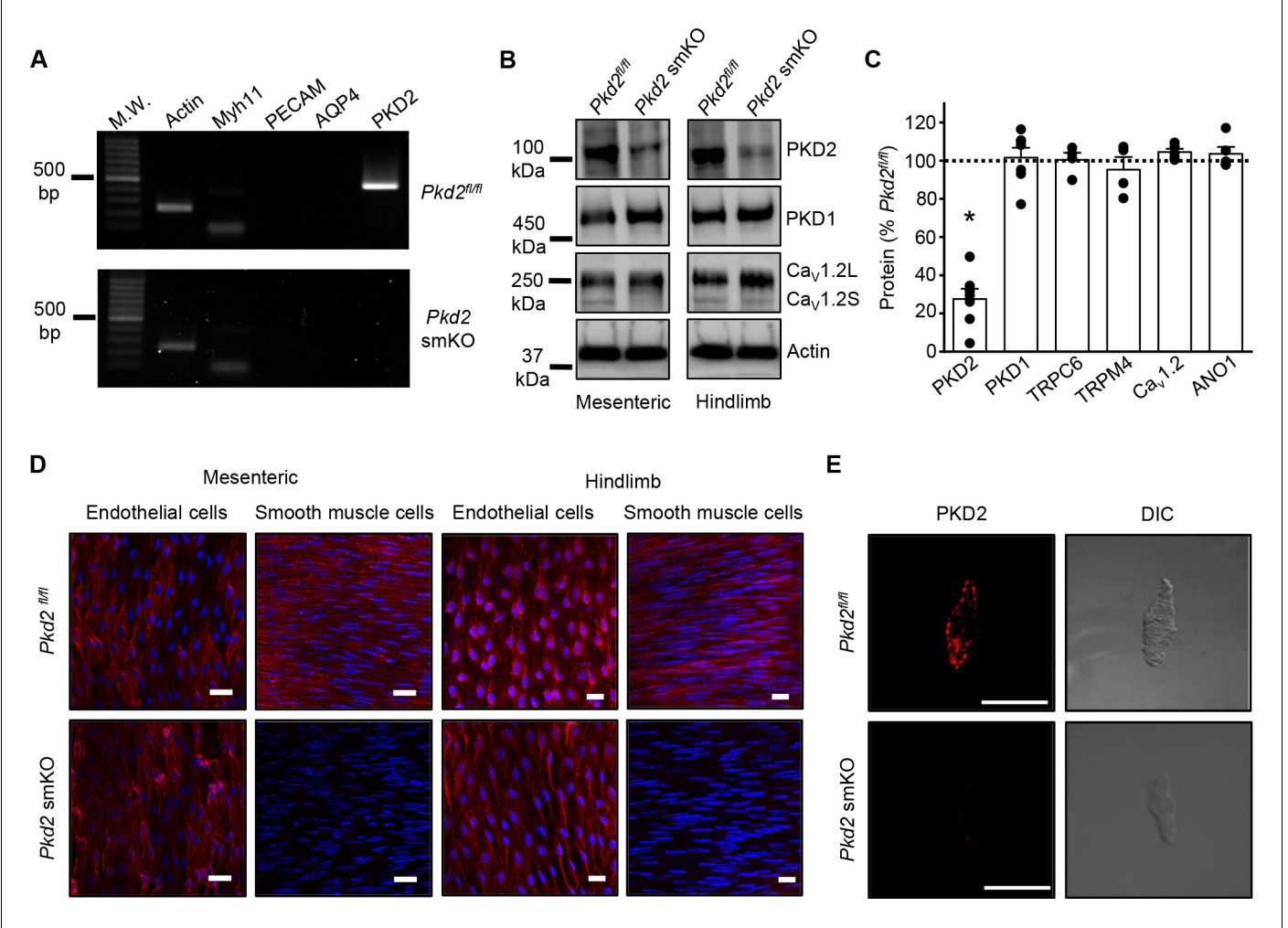

**Figure 1.** Activation of Cre recombinase abolishes PKD2 in arterial myocytes of *Pkd2^fl/fl:myh11cre/ERT2* mice. (**A**) RT-PCR showing the absence of PKD2 transcript in isolated myocytes from tamoxifen-treated *Pkd2^fl/fl:myh11-cre/ERT2* mice. (**B**) Western blots illustrating the effect of tamoxifen-treatment in *Pkd2^fl/fl* and *Pkd2^fl/fl:myh11-cre/ERT2* mice on PKD2, CaV1.2L (full-length CaV1.2) and CaV1.2S (short CaV1.2) proteins in mesenteric and hindlimb arteries. (**C**) Mean data for proteins in mesenteric arteries of tamoxifen-treated *Pkd2^fl/fl:myh11-cre/ERT2* mice when compared to those in tamoxifen-treated *Pkd2^fl/fl* mice. n = 4–7. * indicates p<0.05 versus *Pkd2^fl/fl*. (**D**) *En-face* immunofluorescence imaging illustrating that PKD2 protein (red, Alexa Fluor 555) is abolished in myocytes of mesenteric and hindlimb arteries in tamoxifen-treated *Pkd2^fl/fl:myh11-cre/ERT2* mice (representative of 6 mesenteric and six hindlimb arteries). In contrast, PKD2 protein in endothelial cells is unaltered. Nuclear staining (DAPI) is also shown. Scale bars = 20 μm. (**E**) Confocal and DIC images illustrating that PKD2 protein (Alexa Fluor 555) is abolished in isolated mesenteric artery myocytes of tamoxifen-treated *Pkd2^fl/fl:myh11-cre/ERT2* mice (representative data from 5 *Pkd2^fl/fl* and 5 *Pkd2^fl/fl:myh11-cre/ERT2* mice). Scale bars = 10 μm.

The online version of this article includes the following figure supplement(s) for figure 1:

**Figure supplement 1.** Genotyping of mouse lines.
**Figure supplement 2.** PKD2 protein is lower in aorta and mesenteric and hindlimb arteries from tamoxifen-treated *Pkd2^fl/fl:myh11-cre/ERT2* mice.
**Figure supplement 3.** Several proteins that regulate arterial contractility are unchanged in tamoxifen-treated *Pkd2^fl/fl:myh11-cre/ERT2* mice.

recombination site (*Figure 1A*). Transcripts for PKD2 were amplified by cDNA from arterial myocytes of *Pkd2^fl/fl* mice, but not arterial myocytes of *Pkd2^fl/fl:Myh11-cre/ERT2* mice (*Figure 1A*). These data indicate that tamoxifen induces PKD2 knockout in arterial myocytes of the *Pkd2^fl/fl*:smCre⁺ mice (*Figure 1A*).

Western blotting was performed to quantify proteins in intact arteries of tamoxifen-injected *Pkd2^fl/fl:Myh11-cre/ERT2* and tamoxifen-injected *Pkd2^fl/fl* mice. In mesenteric and hindlimb arteries of *Pkd2^fl/fl:Myh11-cre/ERT2* mice, PKD2 protein was ~25.3% and 32.6%, respectively of that in *Pkd2^fl/fl* controls (*Figure 1B,C*, *Figure 1—figure supplement 2A,B*). In contrast, TRPC6, TRPM4, and ANO1 channels and PKD1, which can form a complex with PKD2 (*Qian et al., 1997*;

*Tsiokas et al., 1997*), were similar between genotypes (*Figure 1C*, *Figure 1—figure supplement 2A,B*). $Ca_V1.2$ protein was similar in mesenteric arteries, but slightly higher (~25.6%) in hindlimb arteries of *Pkd2fl/fl:Myh11-cre/ERT2* mice (*Figure 1B,C*, *Figure 1 – figure supplement B*). In aorta of *Pkd2fl/fl:Myh11-cre/ERT2* mice, PKD2 protein was ~46.8% of that in *Pkd2fl/fl*, whereas $Ca_V1.2$ and PKD1 were similar (*Figure 1—figure supplement 2C and D*). Piezo1, angiotensin II type one receptors and GPR68 have been proposed to act as vascular mechanosensors (*Li et al., 2014*; *Xu et al., 2018*; *Schleifenbaum et al., 2014*; *Blodow et al., 2014*). Piezo1, angiotensin II type one receptors and GPR68 proteins were similar in mesenteric and hindlimb arteries of *Pkd2fl/fl* and *Pkd2fl/fl:Myh11-cre/ERT2* mice (*Figure 1—figure supplement 3*). α-adrenergic receptor subtypes 1A, 1B and 1D were also measured and all were similar in mesenteric and hindlimb arteries of *Pkd2fl/fl* and *Pkd2fl/fl:Myh11-cre/ERT2* mice (*Figure 1—figure supplement 3*).

Immunofluorescence demonstrated that PKD2 protein was present in myocytes of intact arteries and in isolated myocytes of tamoxifen-treated *Pkd2fl/fl* control mice, but absent in myocytes of tamoxifen-treated *Pkd2fl/fl:Myh11-cre/ERT2* mice (*Figure 1D,E*). In contrast, PKD2 protein was present in endothelial cells of both tamoxifen-treated *Pkd2fl/fl* and *Pkd2fl/fl:Myh11-cre/ERT2* mouse arteries (*Figure 1D*). These data indicate that PKD2 detected in Western blots of tamoxifen-treated *Pkd2fl/fl:Myh11-cre/ERT2* mouse arteries is protein present in cell types other than myocytes that would not be targeted by the smooth muscle-specific Cre. These results indicate that PKD2 is expressed in myocytes of mesenteric and hindlimb arteries and aorta and that tamoxifen treatment of *Pkd2fl/fl:Myh11-cre/ERT2* mice selectively abolishes PKD2 expression in myocytes. From this point in the manuscript, tamoxifen-treated *Pkd2fl/fl:Myh11-cre/ERT2* mice will be referred to as *Pkd2* smKO, with tamoxifen-treated *Pkd2fl/fl* mice used as controls in all experiments.

## *Pkd2* smKO mice are hypotensive

Telemetry indicated that diastolic and systolic blood pressures were both lower in *Pkd2* smKO mice than in *Pkd2fl/fl* mice (*Figure 2A and B*). Mean arterial pressure (MAP) was lower in *Pkd2* smKO mice during both day and night cycles, was sustained for days and on average was ~ 22.5% lower in *Pkd2* smKO mice than in *Pkd2fl/fl* mice (*Figure 2C*, *Figure 2—figure supplement 1A*). Locomotion was similar between genotypes, indicating that the lower blood pressure in *Pkd2* smKO mice was not due to inactivity (*Figure 2—figure supplement 1B*). Echocardiography indicated that cardiac output, fractional shortening, ejection fraction and heart rate were similar in *Pkd2* smKO and *Pkd2fl/fl* mice (*Figure 2D*). *Pkd2* smKO kidney glomeruli and tubules were normal and indistinguishable from those of *Pkd2fl/fl* mice (*Figure 2E*). Plasma angiotensin II, aldosterone and ANP and plasma and urine electrolytes were similar in *Pkd2* smKO and *Pkd2fl/fl* mice (*Table 1*, p>0.05 for all). These results demonstrate that arterial myocyte PKD2 channels control systemic blood pressure and that cardiac function and kidney anatomy and function are similar in *Pkd2* smKO and *Pkd2fl/fl* mice.

## Myocyte PKD2 channels are essential for pressure-induced vasoconstriction in hindlimb arteries

Intravascular pressure stimulates vasoconstriction in small, resistance-size arteries. This myogenic response is a major regulator of both regional organ blood flow and systemic blood pressure. To determine whether PKD2 channels contribute to myogenic tone, pressure-induced (20–100 mmHg) vasoconstriction was measured in first-order gastrocnemius arteries. Intravascular pressures greater than 40 mmHg produced increasing levels of constriction in *Pkd2fl/fl* arteries, reaching ~ 20.9% tone at 100 mmHg (*Figure 3A,B*). In contrast, pressure-induced vasoconstriction was robustly attenuated in *Pkd2* smKO gastrocnemius arteries, which developed only ~ 5.4% tone at 100 mmHg or approximately one quarter of that in *Pkd2fl/fl* arteries (*Figure 3A,B*). The passive diameters of *Pkd2fl/fl* and *Pkd2* smKO gastrocnemius arteries were similar (*Figure 3—figure supplement 1A*). Depolarization (60 mM $K^+$) stimulated slightly larger vasoconstriction in *Pkd2* smKO than *Pkd2fl/fl* gastrocnemius arteries (*Figure 3—figure supplement 1B*). These data are consistent with the small increase in $Ca_V1.2$ protein measured in gastrocnemius arteries of *Pkd2* smKO mice and indicate that attenuation of the myogenic response is not due to loss of voltage-dependent $Ca^{2+}$ channel function and that the reduction in systemic blood pressure may stimulate $Ca_V1.2$ expression in hindlimb arteries, but not in mesenteric arteries or aorta (*Figure 1C*, *Figure 1—figure supplement 2B*).

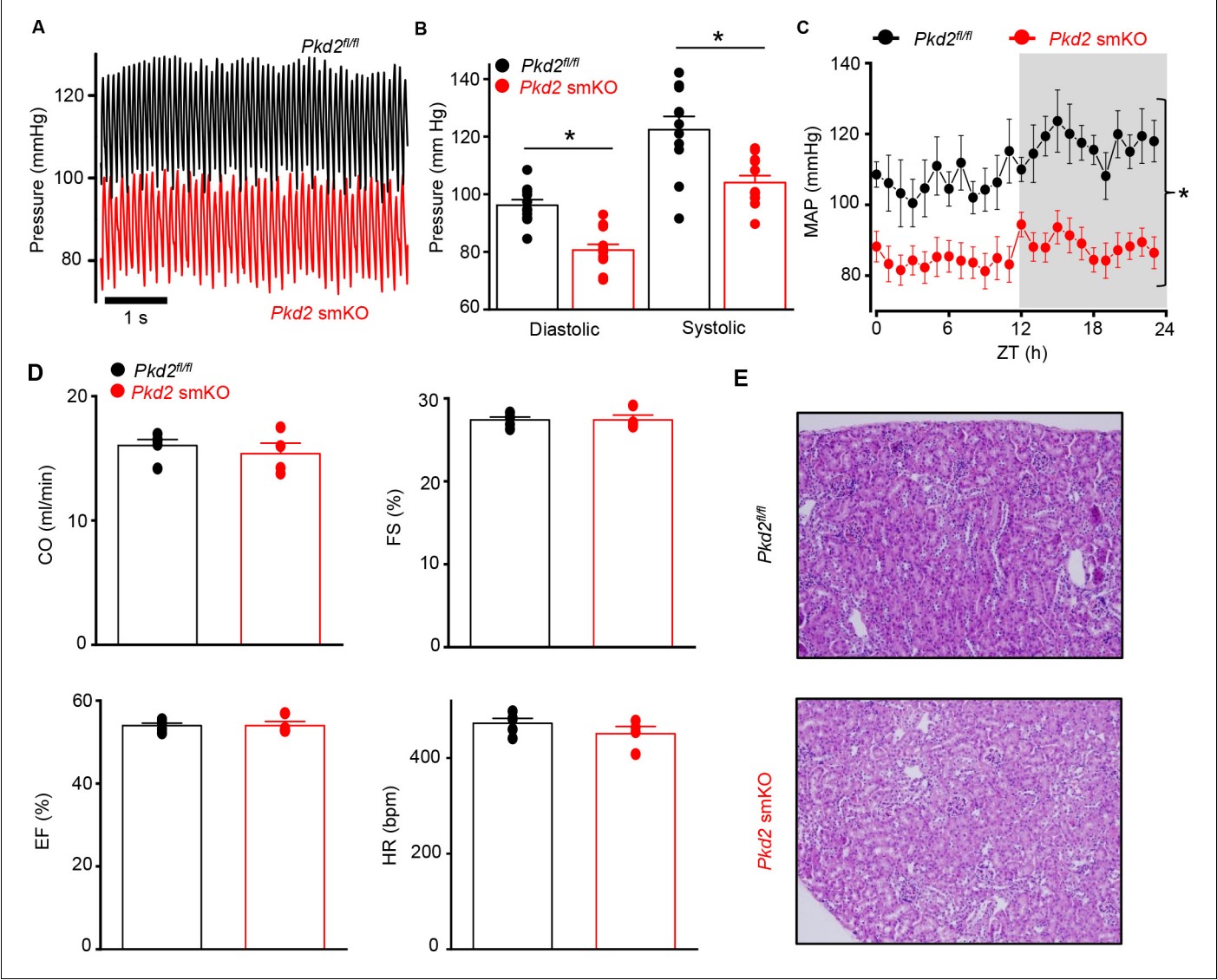

**Figure 2.** *Pkd2* smKO mice are hypotensive with normal cardiac function and renal histology. (A) Original telemetric blood pressure recordings from *Pkd2* smKO and *Pkd2*^fl/fl mice. (B) Mean systolic and diastolic blood pressures in *Pkd2*^fl/fl ($n = 11$) and *Pkd2* smKO ($n = 12$) mice. * indicates $p<0.05$ versus *Pkd2*^fl/fl. (C) Mean arterial blood pressures (MAP) in *Pkd2*^fl/fl ($n = 11$) and *Pkd2* smKO ($n = 12$) mice during day and night (gray) cycles. ZT: Zeitgeber Time. * indicates $p<0.05$ versus *Pkd2*^fl/fl for all data points. (D) Mean echocardiography data. Cardiac output (CO), fractional shortening (FS), ejection fraction (EF) and heart rate (HR). (*Pkd2*^fl/fl, $n = 5$; *Pkd2* smKO mice, $n = 4$). (E) Representative images of H and E stained kidney cortex used for histological assessment ($n = 3$ mice used for for each group).

The online version of this article includes the following figure supplement(s) for figure 2:

**Figure supplement 1.** Lower blood pressure is sustained in *Pkd2* smKO mice.

Sympathetic control of blood pressure occurs in part due to the activation of $\alpha_1$-adrenergic receptors in arterial myocytes. To investigate the contribution of myocyte PKD2 channels to $\alpha_1$-adrenergic receptor-mediated vasoconstriction, responses to phenylephrine, a selective $\alpha_1$-adrenergic receptor agonist, were measured. Phenylephrine stimulated similar vasoconstrictions in pressurized gastrocnemius arteries of *Pkd2*^fl/fl and *Pkd2* smKO mice (*Figure 3—figure supplement 1C*). To determine whether this similar response to receptor agonists was specific to $\alpha_1$-adrenergic receptors, vasoregulation by angiotensin II was measured. Angiotensin II also stimulated similar vasoconstrictions in *Pkd2*^fl/fl and *Pkd2* smKO mouse gastrocnemius arteries (*Figure 3—figure supplement 1D*). To study vasoregulation in intact skeletal muscle, a perfused hindlimb preparation was used.

**Table 1.** Plasma hormones and plasma and urine electrolytes.

|  | *Pkd2*<sup>fl/fl</sup> | *Pkd2* smKO |
|---|---|---|
| Plasma hormones (pg/ml) | | |
| Angiotensin II | 202.5 ± 17.2 (n = 16) | 204.7 ± 12.1 (n = 15) |
| Aldosterone | 341.0 ± 18.2 (n = 16) | 365.6 ± 14.0 (n = 10) |
| ANP | 107.2 ± 9.4 (n = 18) | 118.3 ± 12.3 (n = 18) |
| Plasma electrolytes (mM) | | |
| Na$^+$ | 142.3 ± 1.0 (n = 7) | 152.0 ± 5.5 (n = 6) |
| K$^+$ | 6.4 ± 0.4 (n = 7) | 6.8 ± 0.3 (n = 6) |
| Cl$^-$ | 78.8 ± 0.7 (n = 7) | 84.1 ± 4.3 (n = 6) |
| Urine electrolytes (mM) | | |
| Na$^+$ | 136.9 ± 9.3 (n = 15) | 147.0 ± 16.2 (n = 15) |
| K$^+$ | 563.2 ± 30.2 (n = 15) | 548.6 ± 37.6 (n = 15) |
| Cl$^-$ | 455.4 ± 24.7 (n = 15) | 479.1 ± 42.0 (n = 15) |

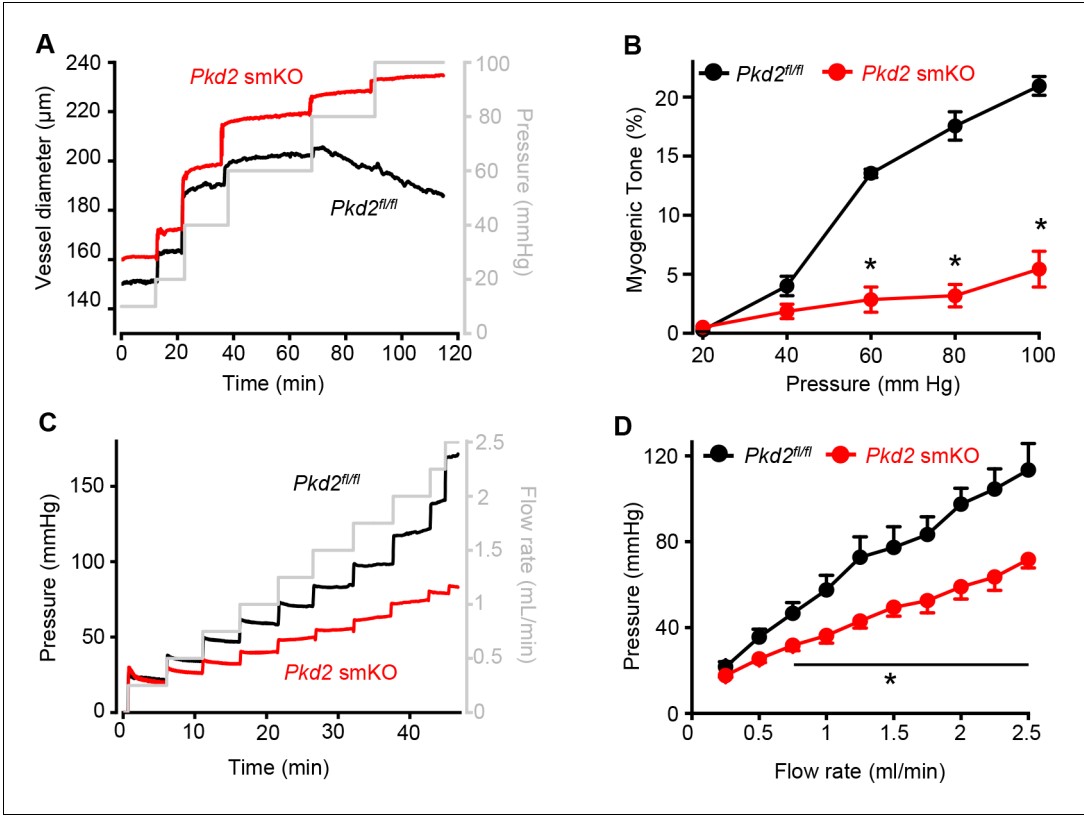

**Figure 3.** Pressure-induced vasoconstriction is attenuated in *Pkd2* smKO mouse hindlimb arteries. (**A**) Representative traces illustrating diameter responses to intravascular pressure in gastrocnemius arteries of *Pkd2*<sup>fl/fl</sup> and *Pkd2* smKO mice. (**B**) Mean data for myogenic tone in gastrocnemius arteries (*Pkd2*<sup>fl/fl</sup>, n = 5; *Pkd2* smKO, n = 6). * indicates p<0.05 versus *Pkd2*<sup>fl/fl</sup>. (**C**) Representative traces illustrating hindlimb perfusion pressure in response to increasing flow. (**D**): Mean data for hindlimb perfusion pressure (*Pkd2*<sup>fl/fl</sup>, n = 6; *Pkd2* smKO, n = 4). * indicates p<0.05 versus *Pkd2*<sup>fl/fl</sup>.

The online version of this article includes the following figure supplement(s) for figure 3:

**Figure supplement 1.** Myocyte PKD2 knockout does not alter phenylephrine or angiotensin II-induced vasoconstriction in hindlimb arteries.

Stepwise increases in intravascular flow produced lower pressures in hindlimbs of *Pkd2* smKO mice than in those of *Pkd2*[fl/fl] mice (*Figure 3C,D*). For example, at a flow rate of at 2.5 ml/min the mean pressure in hindlimbs of *Pkd2* smKO mice were ~ 63.2% of those in *Pkd2*[fl/fl] mice (*Figure 3C,D*). In contrast, at constant flow, phenylephrine similarly increased pressure in hindlimbs of *Pkd2* smKO and *Pkd2*[fl/fl] mice (*Figure 3—figure supplement 1E*). These data suggest that arterial myocyte PKD2 channels are essential for pressure-induced vasoconstriction, but not phenylephrine- or angiotensin II-induced vasoconstriction, in hindlimb arteries.

## Pressure-induced membrane depolarization requires myocyte PKD2 channels in hindlimb arteries

To investigate the mechanism(s) by which myocyte PKD2 channels regulate contractility, membrane potential was measured in pressurized hindlimb arteries using glass microelectrodes. At 10 mmHg, the mean membrane potential of *Pkd2*[fl/fl] and *Pkd2* smKO arteries were similar at ~ −59.6 and −58.5 mV, respectively (*Figure 4A,B*). Increasing intravascular pressure to 100 mmHg depolarized *Pkd2*[fl/fl] arteries by ~ 22.5 mV, but did not alter the membrane potential of arteries from *Pkd2* smKO mice (*Figure 4A,B*). In contrast, phenylephrine depolarized arteries to similar membrane potentials in both genotypes (*Figure 4A,B*). These data suggest that pressure activates PKD2 channels in myocytes of hindlimb arteries, leading to depolarization and vasoconstriction.

## Swelling activates PKD2 channels in hindlimb artery myocytes

The contribution of PKD2 channels to mechanosensitive currents was investigated in hindlimb artery myocytes. Recent evidence indicates that recombinant PKD2 generates voltage-dependent, outwardly rectifying currents and is primarily a $Na^+$-permeant channel under physiological membrane potentials and ionic gradients (*Shen et al., 2016*; *Gonzalez-Perrett et al., 2002*). Whole cell $I_{Cat}$ was recorded using the whole-cell patch-clamp configuration with symmetrical $Na^+$ solutions. In *Pkd2*[fl/fl] myocytes, reducing bath solution osmolarity from 300 to 250 mOsm caused cell swelling and activated an outwardly rectifying $I_{Cat}$ that increased $2.3 \pm 0.3$ and $3.0 \pm 0.7$ fold at −100 and + 100 mV, respectively ($p<0.05$, n = 5, *Figure 4C*). Swelling-activated $I_{Cat}$ was inhibited by $Gd^{3+}$, a non-selective cation channel blocker (*Figure 4C,D,E*). The mean reversal potential for $I_{Cat}$ was similar in isotonic (−0.2 ± 1.3 mV) and hypotonic solutions (0.1 ± 3.8 mV; $p>0.05$), suggesting that swelling activated a $I_{Na}$. To test this conclusion experimentally, $Na^+$ substitution experiments were performed. In a hypotonic bath solution, a reduction in bath [$Na^+$] from 115 to 40 mM shifted the $E_{rev}$ to −14.6 ± 0.5 mV in *Pkd2*[fl/fl] myocytes (n = 5, $p<0.05$ versus $E_{rev}$ in 250 mOsm) (*Figure 4F*). When adjusted for the liquid junction potential caused by the solution change (+6.9 mV), the corrected $E_{rev}$ is −21.5 ± 0.5 mV, which is similar to the calculated $E_{Na}$ (−24.2 mV). Reducing bath $Na^+$ also decreased inward current and increased outward current. In contrast, swelling did not activate a $I_{Cat}$ ($p<0.05$, n = 5), nor did $Gd^{3+}$ alter $I_{Cat}$ when applied in hypotonic bath solution, in hindlimb artery myocytes of *Pkd2* smKO mice (*Figure 4C,D,E*). In further contrast to the differential effects of cell swelling, phenylephrine activated similar amplitude $I_{Cat}$s in hindlimb artery myocytes of *Pkd2*[fl/fl] and *Pkd2* smKO mice (*Figure 4—figure supplement 1*). These data indicate that cell swelling activates PKD2 channels, leading to a $Na^+$ current in hindlimb artery myocytes, whereas phenylephrine does not activate PKD2 channels in this cell type.

## Myocyte PKD2 channels contribute to phenylephrine-induced vasoconstriction in mesenteric arteries

Mesenteric arteries were studied to determine whether myocyte PKD2 channels control contractility in another arterial bed that is a major regulator of systemic blood pressure. In contrast to the robust attenuation of myogenic vasoconstriction in gastrocnemius arteries of *Pkd2* smKO mice, pressure- and depolarization-induced vasoconstriction was similar in mesenteric arteries of *Pkd2* smKO and *Pkd2*[fl/fl] mice (*Figure 5A*, *Figure 5—figure supplement 1B*). Myogenic tone was similar regardless of whether third, fourth, or fifth-order *Pkd2* smKO and *Pkd2*[fl/fl] mouse mesenteric arteries were studied (*Figure 5—figure supplement 1A*). Passive diameter of mesenteric arteries was not altered by myocyte PKD2 knockout (*Figure 3—figure supplement 1A*). The differential contribution of myocyte PKD2 channels to pressure-induced vasoconstriction in gastrocnemius and mesenteric arteries

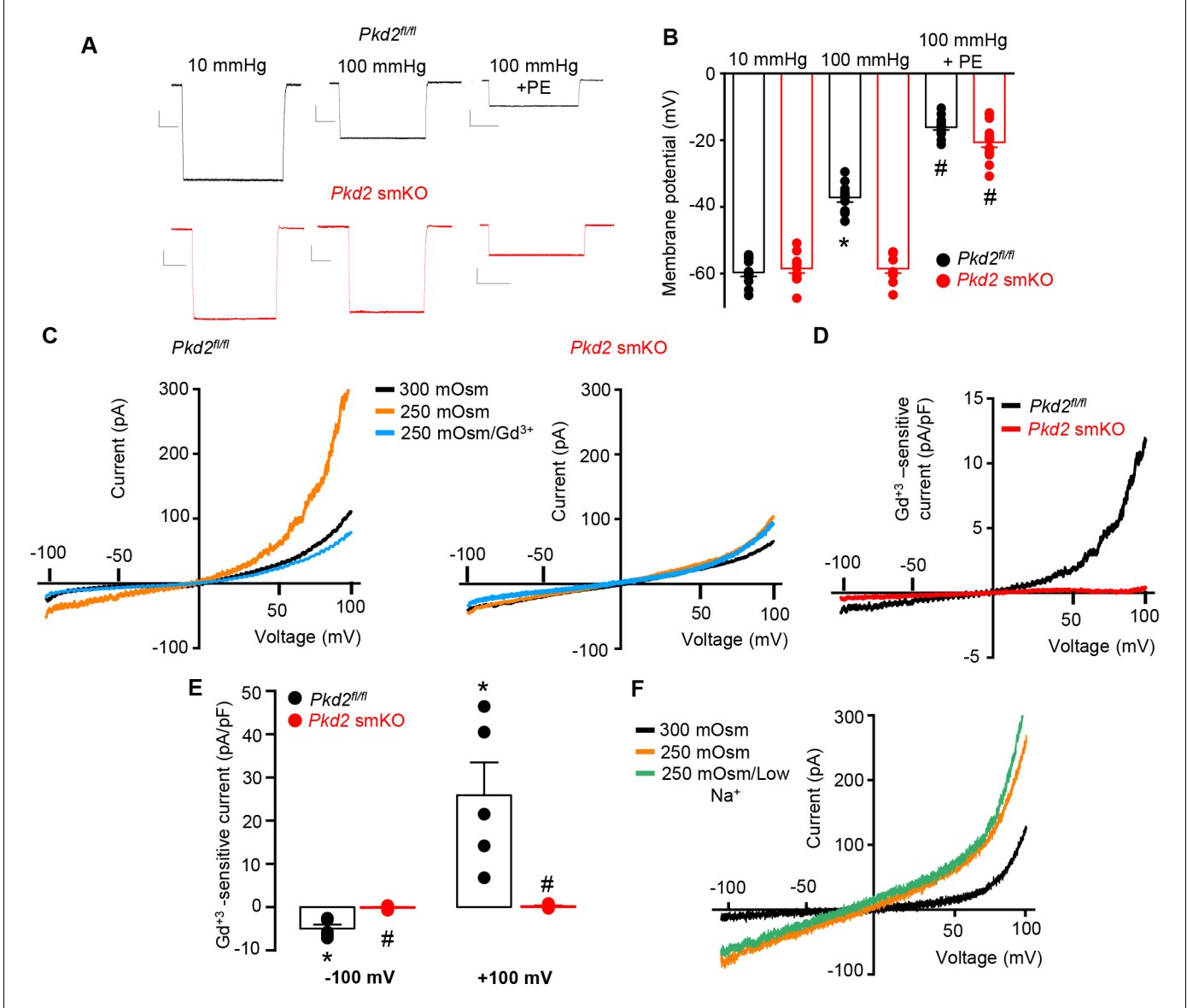

**Figure 4.** PKD2 channels contribute to pressure-induced hindlimb artery depolarization and swelling-activated Na + currents in hindlimb artery myocytes. (**A**) Representative traces of microelectrode impalements under indicated conditions illustrating that pressure-induced depolarization is attenuated in gastrocnemius arteries of *Pkd2* smKO mice. Phenylephrine (PE) = 1 μM. Scale bars: Y = 10 mV, X = 20 s. (**B**) Mean data for membrane potential recordings in pressurized hindlimb arteries in the absence or presence of PE (*Pkd2^fl/fl*: 10 mmHg, n = 11; 100 mmHg, n = 10; 100 mmHg + PE, n = 13 and *Pkd2* smKO: 10 mmHg, n = 11; 100 mmHg, n = 10; 100 mmHg + PE, n = 14). * indicates p<0.05 versus 10 mmHg in *Pkd2^fl/fl*. # indicates p<0.05 versus 100 mmHg in the same genotype. (**C**) Representative ICats recorded between −100 and +100 mV in isotonic (300 mOsm), hypotonic (250 mOsm) and hypotonic bath solution with Gd^{3+} (100 μM) in the same *Pkd2^fl/fl* and *Pkd2* smKO mouse hindlimb artery myocytes. (**D**) Representative I-V relationships of Gd^{+3}-sensitive Icats activated by hypotonic solution in *Pkd2^fl/fl* and *Pkd2* smKO hindlimb myocytes. (**E**) Mean data for Gd^{3+}-sensitive ICats recorded in hypotonic solution in *Pkd2^fl/fl* and *Pkd2* smKO myocytes (n = 5 for each). * indicates p<0.05 versus 250 mOsm, # p<0.05 versus *Pkd2^fl/fl*. (**F**) Representative I-V relationships between −100 and +100 mV in isotonic (300 mOsm), hypotonic (250 mOsm) and hypotonic bath solution with low (40 mM) Na^+ in the same *Pkd2^fl/fl* mouse hindlimb artery myocyte.

The online version of this article includes the following figure supplement(s) for figure 4:

**Figure supplement 1.** PKD2 knockout does not alter phenylephrine (PE)-activated ICat in isolated hindlimb artery myocytes.

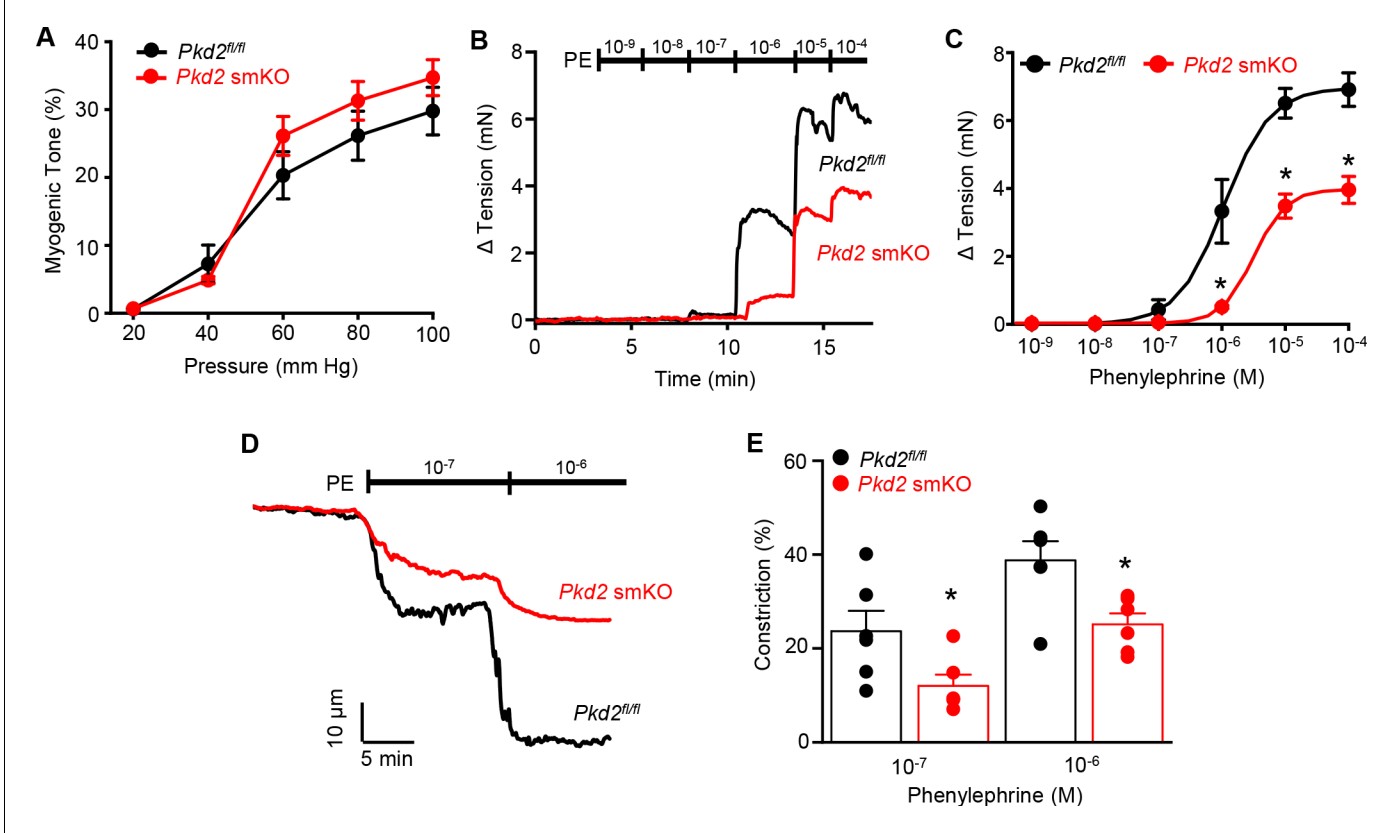

**Figure 5.** Pressure-induced vasoconstriction is unaltered, whereas phenylephrine-induced vasoconstriction is attenuated, in mesenteric arteries of *Pkd*2 smKO mice. (A) Mean vasoconstriction over a range of pressures in resistance-size mesenteric arteries from *Pkd2*$^{fl/fl}$ (n = 7) and *Pkd*2 smKO (n = 9) mice. (B) Original recordings of concentration-dependent, phenylephrine (PE)-induced contraction in mesenteric artery rings. (C) Mean PE-induced contraction (*Pkd2*$^{fl/fl}$, n = 5; *Pkd*2 smKO, n = 6; *p<0.05 versus *Pkd2*$^{fl/fl}$). (D) Representative phenylephrine-induced vasoconstriction in pressurized (80 mmHg) fifth-order mesenteric arteries. (E) Mean PE-induced vasoconstriction in pressurized (80 mmHg) fourth-and fifth-order mesenteric arteries (*Pkd2*$^{fl/fl}$, n = 6; *Pkd*2 smKO, n = 6; *p<0.05 versus *Pkd2*$^{fl/fl}$ at the same PE concentration).

The online version of this article includes the following figure supplement(s) for figure 5:

**Figure supplement 1.** Myocyte PKD2 knockout attenuates phenylephrine-induced vasoconstriction, but does not alter pressure or angiotensin II-induced vasoconstriction in mesenteric arteries.

was not due to size, as passive diameters of first-order gastrocnemius and third-order mesenteric arteries were similar (*Figure 3—figure supplement 1A*).

The splanchnic circulation receives considerable sympathetic innervation. To investigate the contribution of myocyte PKD2 channels to $\alpha_1$-adrenoceptor-mediated responses, phenylephrine-induced vasoconstriction was measured. Phenylephrine-induced isometric contractions in mesenteric artery (first- and second-order) rings of *Pkd*2 smKO mice were smaller than those in *Pkd2*$^{fl/fl}$ controls (*Figure 5B,C*). For instance, with 1 µM phenylephrine, contractions in *Pkd*2 smKO arteries were ~ 15.3% of those in *Pkd2*$^{fl/fl}$ arteries, with maximal phenylephrine-induced contraction ~ 57.3% of that in *Pkd2*$^{fl/fl}$ arteries (*Figure 5B,C*). The mean concentration of phenylephrine-induced half-maximal contraction (EC$_{50}$, µM) was slightly higher in *Pkd*2 smKO (2.4 ± 0.2) than *Pkd2*$^{fl/fl}$ (1.1 ± 0.3) arteries (p<0.05, *Figure 5B,C*). In pressurized (80 mmHg) mesenteric (fourth- and fifth-order) arteries of *Pkd*2 smKO mice, phenylephrine-induced vasoconstrictions were between ~ 50.9% and 64.8% of those in *Pkd2*$^{fl/fl}$ controls (*Figure 5D,E*). Similar results were obtained with endothelium-denuded mesenteric arteries, indicating that attenuated vasoconstriction to phenylephrine was due to loss of PKD2 in myocytes (*Figure 5—figure supplement 1C and D*). In contrast to attenuated phenylephrine-mediated vasoconstriction, angiotensin II-induced constriction was not different between *Pkd2*$^{fl/fl}$ and *Pkd*2 smKO mesenteric arteries (*Figure 5—figure supplement 1E*). These data indicate that arterial myocyte PKD2 channels are activated by distinct vasoconstrictor stimuli in arteries of

different tissues, contributing to $\alpha_1$-adrenoceptor-mediated vasoconstriction in mesenteric arteries and essential for the myogenic response in hindlimb arteries.

## $\alpha_1$-adrenergic receptors stimulate myocyte PKD2 channels, leading to membrane depolarization in mesenteric arteries

At an intravascular pressure of 10 mmHg, the membrane potential of $Pkd2^{fl/fl}$ and $Pkd2$ smKO mesenteric arteries was similar (*Figure 6A,B*). An increase in pressure to 80 mmHg stimulated a similar depolarization in mesenteric arteries of $Pkd2^{fl/fl}$ and $Pkd2$ smKO mice, to ~39.7 and 36.7 mV, respectively (*Figure 6A,B*). Phenylephrine application further depolarized $Pkd2^{fl/fl}$ mesenteric arteries by ~19.9 mV, but did not change the membrane potential of $Pkd2$ smKO arteries (*Figure 6A,B*).

To examine phenylephrine-regulation of $I_{Cat}$ in isolated mesenteric artery myocytes, the whole-cell patch-clamp configuration was used with symmetrical 140 mM NaCl. In $Pkd2^{fl/fl}$ cells, phenylephrine activated outwardly rectifying $I_{Cat}$, increasing density ~ 1.43 and 1.57-fold at $-100$ mV and $+ 100$ mV, respectively (*Figure 6C,D*). In the presence of phenylephrine, a reduction in bath $[Na]_o$ from 140 to 40 mM shifted the $E_{rev}$ from $0.5 \pm 0.5$ to $-23.2 \pm 1.6$ mV in $Pkd2^{fl/fl}$ myocytes

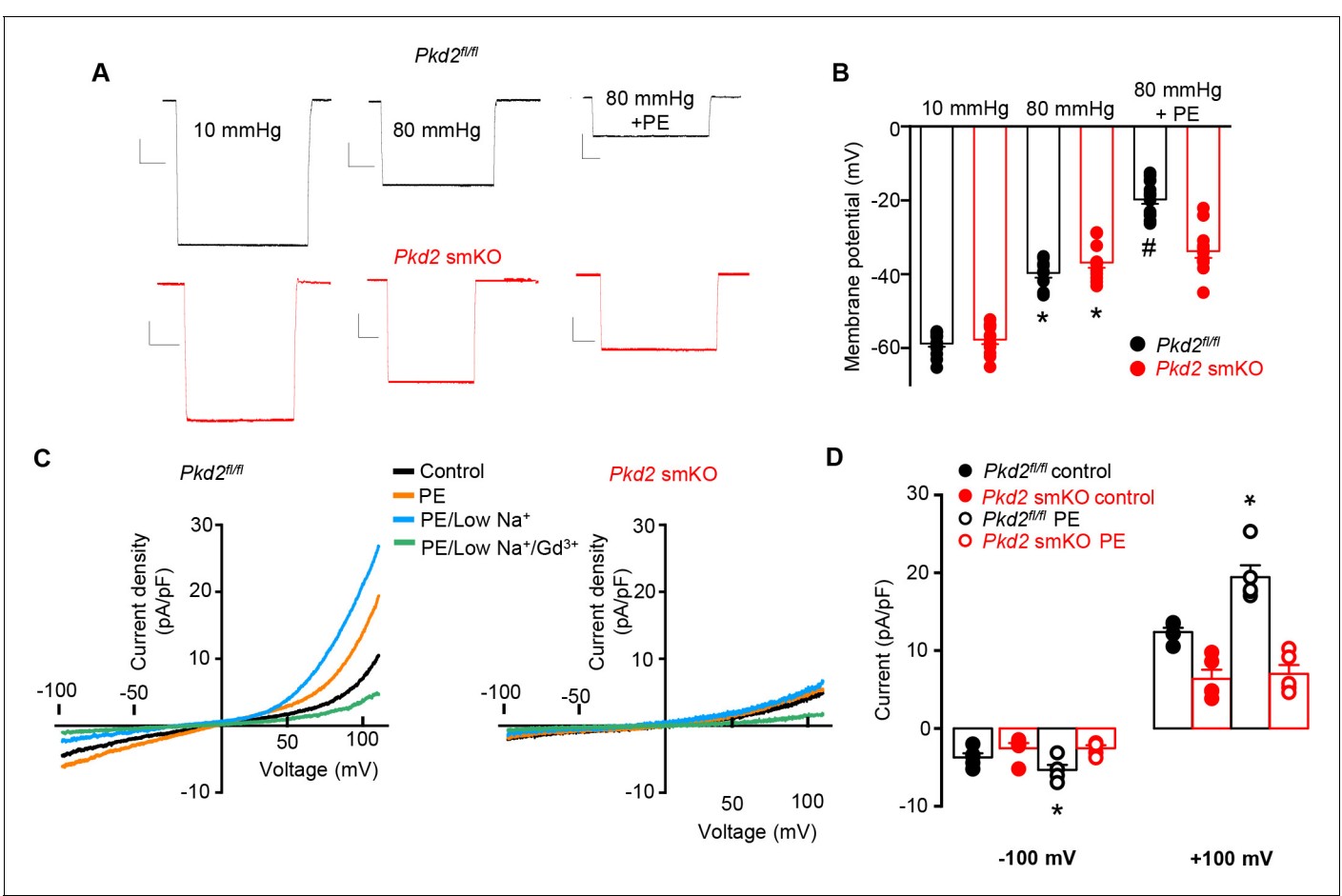

**Figure 6.** PKD2 channels contribute to phenylephrine-induced mesenteric artery depolarization and INa in mesenteric artery myocytes. (**A**) Representative traces of microelectrode impalements illustrating that phenylephrine (PE, 1 μM)-induced depolarization is attenuated in mesenteric arteries of $Pkd2$ smKO mice. Scale bars: Y = 10 mV, X = 20 s. (**B**) Mean membrane potential recordings in pressurized (10 and 80 mmHg) mesenteric arteries and in PE at 80 mmHg ($Pkd2^{fl/fl}$: 10 mmHg, n = 13; 80 mmHg, n = 9; 80 mmHg + PE, n = 15. $Pkd2$ smKO: 10 mmHg, n = 11; 80 mmHg, n = 12; 80 mmHg + PE, n = 12). *p<0.05 versus 10 mmHg in the same genotype. # p<0.05 versus 80 mmHg in the same genotype. (**C**) Original current recordings obtained between $-100$ and +100 mV in the same $Pkd2^{fl/fl}$ and $Pkd2$ smKO myocytes in control, PE (10 μM), low Na+ (40 mM)+PE and low Na+ (40 mM)+PE + Gd3+ (100 μM). (**D**) Mean paired data ($Pkd2^{fl/fl}$, n = 5; $Pkd2$ smKO, n = 5; *p<0.05 versus control in the same genotype).

The online version of this article includes the following figure supplement(s) for figure 6:

**Figure supplement 1.** PKD2 knockout does not alter swelling-activated Icat in isolated mesenteric artery myocytes.

(n = 5, p<0.05 versus PE/low Na$^+$). When corrected for the liquid junction potential (+5.6 mV) caused by the change in bath [Na]$_o$, this E$_{rev}$ is $-28.8 \pm 1.6$ mV, which is similar to calculated E$_{Na}$ ($-32.8$ mV) under these conditions. Reducing bath Na$^+$ also reduced inward current and increased outward current. The subsequent addition of Gd$^{3+}$ reduced current density to ~ 44.9% and 20.7% of that in phenylephrine/40 mM Na$^+$at $-100$ and + 100 mV, respectively (n = 5, p<0.05, *Figure 6C*). In contrast to effects in *Pkd2$^{fl/fl}$* myocytes, phenylephrine did not alter I$_{Cat}$ in *Pkd2* smKO myocytes (*Figure 6C,D*). In further contrast to the differential effects of phenylephrine, swelling activated similar amplitude I$_{Cat}$ in mesenteric artery myocytes of *Pkd2$^{fl/fl}$* and *Pkd2* smKO mice (*Figure 6—figure supplement 1*). These data indicate that phenylephrine activates PKD2 channels, leading to a I$_{Na}$ in mesenteric artery myocytes, whereas cell swelling does not activate PKD2 channels in mesenteric artery myocytes.

## PKD2 channel knockout in arterial myocytes attenuates hypertension

We tested the hypothesis that arterial myocyte PKD2 channels are associated with the increase in blood pressure during hypertension. Angiotensin II-infusion is a well-established method to produce a stable elevation in mean arterial pressure. Blood pressure was measured following implantation of subcutaneous osmotic minipumps that infused angiotensin II or saline in *Pkd2* smKO and *Pkd2$^{fl/fl}$* mice. Angiotensin II steadily increased MAP to a plateau of ~155.6 mmHg in *Pkd2$^{fl/fl}$* mice and to ~ 134.6 mmHg in *Pkd2* smKO mice (*Figure 7A*). The angiotensin II-induced increase in MAP (ΔMAP) was ~ 25.6% smaller in *Pkd2* smKO than *Pkd2$^{fl/fl}$* mice (*Figure 7A*). Saline infusion did not alter blood pressure in either *Pkd2* smKO or *Pkd2$^{fl/fl}$* mice. These data indicate that myocyte PKD2 channel knockout reduces both the absolute systemic blood pressure and the increase in blood pressure during hypertension.

## Hypertension is associated with an upregulation of both total and surface PKD2 proteins in systemic arteries

To investigate mechanisms by which arterial myocyte PKD2 channels may be associated with an increase in blood pressure during hypertension, total and surface proteins were measured in arteries. PKD2 total protein in mesenteric and hindlimb arteries of angiotensin II-induced hypertensive mice were ~ 149.2% and 143.4%, respectively, of those in normotensive mice (*Figure 7B,C*). Arterial biotinylation revealed that surface PKD2 protein in mesenteric and hindlimb arteries of hypertensive mice were also ~ 150.3% and 145.9% of those in controls (*Figure 7D,E*). In contrast, cellular distribution of PKD2 was similar in mesenteric and hindlimb arteries of normotensive and hypertensive mice, with ~ 85% of protein located in the plasma membrane (*Figure 7F*). These data indicate that during hypertension, an increase in total PKD2 protein leads to an increase in the abundance of surface PKD2 channels.

## Myocyte-specific PKD2 channel knockout induces vasodilation and prevents arterial remodeling during hypertension

To test the hypothesis that the reduction in systemic blood pressure in *Pkd2* smKO mice during hypertension was due to vasodilation, the contractility of pressurized (80 mmHg) mesenteric arteries was measured using myography. Phenylephrine-induced vasoconstrictions in angiotensin II-treated *Pkd2* smKO mouse arteries were between ~67.6% and 71.1% of those in *Pkd2$^{fl/fl}$* arteries (*Figure 8A*). In contrast, myogenic tone was similar in arteries of angiotensin II-treated *Pkd2* smKO and *Pkd2$^{fl/fl}$* mice (*Figure 8B*). These results indicate that knockout of arterial myocyte PKD2 channels attenuates phenylephrine-induced vasoconstriction during hypertension. These data also suggest that hypertension does not promote the emergence of a mechanism by which PKD2 channels contribute to the myogenic response in mesenteric arteries.

Hypertension is associated with inward remodeling of vasculature (*Schiffrin, 2012*). To investigate the involvement of myocyte PKD2 channels in pathological remodeling, arterial sections from angiotensin II-treated *Pkd2* smKO and *Pkd2$^{fl/fl}$* mice were imaged and analyzed. In saline-treated *Pkd2* smKO and *Pkd2$^{fl/fl}$* mice, the wall-to-lumen ratio of mesenteric arteries was similar (*Figure 8C,D*). Angiotensin II infusion increased the artery wall-to-lumen ratio ~2.9 fold in *Pkd2$^{fl/fl}$* mice, but only 1.3-fold in *Pkd2* smKO mice, or ~89.6% less (*Figure 8C,D*). These data suggest that PKD2 knockout in myocytes attenuates arterial remodeling during hypertension.

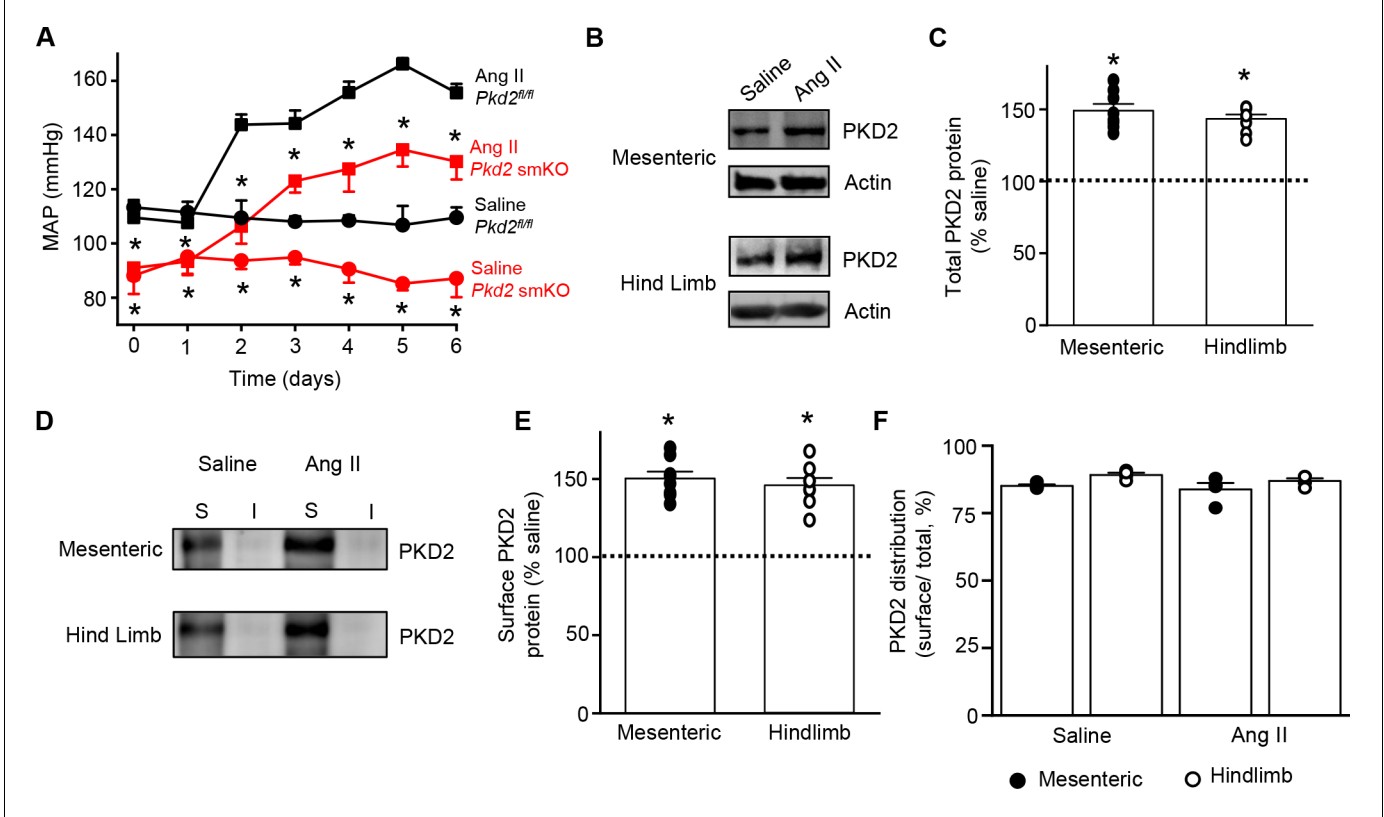

**Figure 7.** Angiotensin II-induced hypertension is attenuated in *Pkd2* smKO mice. (A) Telemetric blood pressure time course showing the development of angiotensin II-induced hypertension in *Pkd2^{fl/fl}* (n = 6) and *Pkd2* smKO mice (n = 9). Osmotic minipumps containing either saline or angiotensin II were implanted one day prior to day 0. * indicates p<0.05 versus *Pkd2^{fl/fl}* in the same condition. (B) Western blots illustrating total PKD2 protein in mesenteric and hindlimb arteries of saline-and angiotensin II-treated control mice. (C) Mean total PKD2 protein in mesenteric and hindlimb arteries of angiotensin II-treated mice compared to saline control (n = 8 for each group). * indicates p<0.05 versus saline in the same arterial preparation. (D) Western blots showing surface and intracellular PKD2 protein in arteries of saline-and angiotensin II-treated mice. (E) Mean surface PKD2 protein in mesenteric and hindlimb arteries of angiotensin II-treated mice compared to saline control (n = 8 for each group). * indicates p<0.05 versus saline in the same arterial preparation. (F) Mean data illustrating the percentage of total PKD2 located at the surface in mesenteric and hindlimb arteries of saline-and angiotensin II-treated mice (n = 4 for each group).

## Discussion

Previous studies performed in vitro have suggested that arterial myocytes express several different TRP channels, but in vivo physiological functions of these proteins are unclear. Here, we generated an inducible, smooth muscle-specific knockout of a TRP channel, PKD2, to investigate blood pressure regulation by this protein. We show that tamoxifen-induced, smooth muscle-specific PKD2 knockout dilates resistance-size systemic arteries and reduces blood pressure. Data indicate that heterogeneous vasoconstrictor stimuli activate PKD2 channels in arterial myocytes of different tissues. PKD2 channel activation leads to a $I_{Na}$ in myocytes, which induces membrane depolarization and vasoconstriction. Furthermore, we show that hypertension is associated with an increase in plasma membrane-resident PKD2 channels. PKD2 channel knockout in myocytes of hypertensive mice caused vasodilation, prevented arterial remodeling and lowered systemic blood pressure. In summary, using an inducible, conditional knockout model we show that arterial myocyte PKD2 channels increase physiological systemic blood pressure. We also show that arterial myocyte PKD2 channels are upregulated during hypertension and genetic knockout reduces high blood pressure.

The regulation of blood pressure by arterial myocyte TRP proteins and involvement in hypertension are poorly understood (*Earley and Brayden, 2015*). This lack of knowledge is due to several primary factors. First, it is unclear which TRP channel family members are expressed and functional in myocytes of resistance-size systemic arteries that control blood pressure. The identification of a

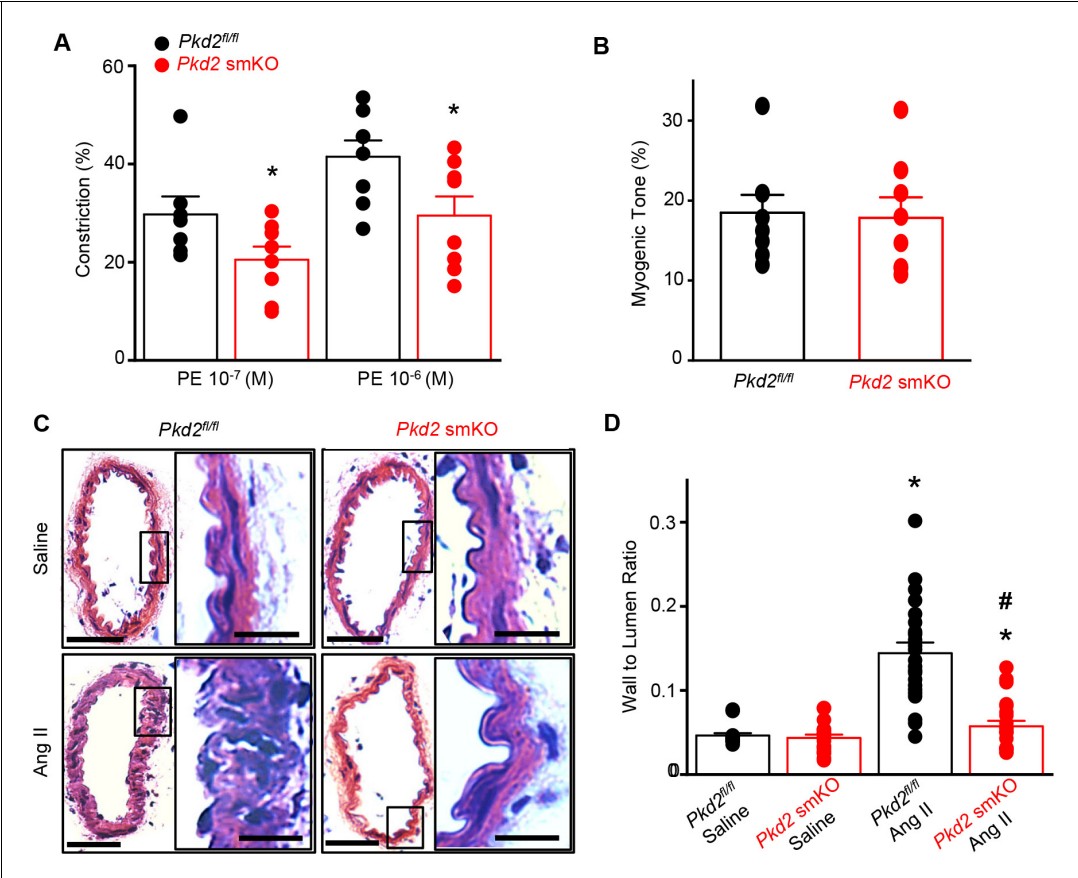

**Figure 8.** Arterial myocyte PKD2 knockout attenuates vasoconstriction and arterial wall remodeling during hypertension. (**A**) Mean phenylephrine-induced vasoconstriction in pressurized (80 mmHg) mesenteric arteries from angiotensin II-treated mice (*Pkd2^fl/fl*, n = 7–8; *Pkd2* smKO, n = 8).

TRP channel in blood vessels that do not regulate systemic blood pressure, such as aorta and cerebral arteries, does not necessarily indicate that the same protein is present in arteries that do control blood pressure (*Earley and Brayden, 2015*). Second, TRP channel expression has often been reported in myocytes that had undergone cell culture or in whole vasculature that contains many different cell types. These studies create uncertainty as to which TRP proteins are present specifically in contractile myocytes of resistance-size systemic arteries (*Earley and Brayden, 2015*). Third, the lack of specific modulators of individual TRP channel subfamily members has significantly hampered studies aimed at identifying in vitro and in vivo functions of these proteins. Fourth, TRP channels are expressed in many different cell types other than arterial myocytes. Blood pressure measurements in global, non-inducible TRP knockout mice generated contrasting results to those anticipated. Myocyte TRPM4 channel activation leads to vasoconstriction in isolated cerebral arteries, but global, constitutive knockout of TRPM4 elevated blood pressure in mice (*Mathar et al., 2010*; *Earley et al., 2004*). Arterial myocyte TRPC6 channel activation contributes to myogenic tone in cerebral arteries, but global, constitutive TRPC6 knockout caused vasoconstriction and elevated blood pressure in mice (*Dietrich et al., 2005*). TRPV4 channel activation in arterial myocytes leads to vasodilation in isolated vessels, but blood pressure in TRPV4^-/- mice was the same or lower than controls (*Nishijima et al., 2014*; *Earley et al., 2009*). Alternative approaches, such as the inducible, cell-specific knockout model developed here, are valuable to advance knowledge of blood pressure regulation by arterial myocyte TRP channels.

Previous studies performed in vitro that investigated vasoregulation by PKD2 channels produced a variety of different findings. RNAi-mediated knockdown of PKD2 inhibited swelling-activated non-selective cation currents ($I_{Cat}$) in rat cerebral artery myocytes and reduced myogenic tone in pressurized cerebral arteries (*Narayanan et al., 2013*). In contrast, PKD2 siRNA did not alter myogenic tone

in mesenteric arteries of wild-type mice, but increased both stretch-activated cation currents and myogenic vasoconstriction in arteries of constitutive, myocyte PKD1 knockout mice (*Sharif-Naeini et al., 2009*). From these results, the authors suggested that myocyte PKD2 channels inhibit mesenteric vasoconstriction (*Sharif-Naeini et al., 2009*). Global knockout of PKD2 is embryonic-lethal in mice due to cardiovascular and renal defects, prohibiting study of the loss of protein in all tissues on vascular function (*Wu et al., 2000*; *Boulter et al., 2001*). In global PKD2$^{+/-}$ mice, an increase in actin and myosin expression lead to larger phenylephrine-induced contraction in aorta and mesenteric arteries (*Qian et al., 2007*; *Du et al., 2010*). Here, using an inducible cell-specific knockout model, we show that swelling, a mechanical stimulus, activates PKD2 currents in myocytes of skeletal muscle arteries, whereas phenylephrine activates PKD2 currents in myocytes of mesenteric arteries. Regardless of the stimulus involved, PKD2 activation leads to a Na$^+$ current, membrane depolarization and vasoconstriction in both artery types. PKD2 appears to be the second TRP protein to have been described in in arterial myocytes of skeletal muscle arteries, TRPV1 being the other (*Kark et al., 2008*).

Few studies have measured plasma membrane currents through either recombinant PKD2 channels or PKD2 channels expressed in native cell types. PKD2 channels have been observed at the cell surface, with evidence suggesting that PKD1 is required for PKD2 translocation, although the ability of PKD1 to perform this function has been questioned (*Shen et al., 2016*; *Hanaoka et al., 2000*). Here, using arterial biotinylation we show that ~85% of PKD2 protein is located in the plasma membrane in both mesenteric and hindlimb arteries, consistent with our previous data in cerebral arteries (*Narayanan et al., 2013*). PKD2 ion permeability has also been a matter of debate. Recombinant PKD2 was initially shown to both conduct Ca$^{2+}$ and to be inhibited by Ca$^{2+}$ (*Hanaoka et al., 2000*; *Cai et al., 2004*). To increase surface-trafficking of recombinant PKD2, a recent study generated a chimera by replacing the pore of PKD2-L1, a related channel that readily traffics to the plasma membrane, with the PKD2 channel pore (*Shen et al., 2016*). This channel, which contained the PKD2 channel filter, generated outwardly-rectifying whole cell currents and was more selective for Na$^+$ and K$^+$ than Ca$^{2+}$, with Ca$^{2+}$ only able to permeate in an inward direction (*Shen et al., 2016*). We also show that PKD2 channels in hindlimb and mesenteric artery myocytes are outwardly rectifying and permeant to Na$^+$. At physiological arterial potentials of ~$-60$ to $-40$ mV, PKD2 channel opening would produce an inward Na$^+$ current, leading to myocyte membrane depolarization and voltage-dependent Ca$^{2+}$ (Ca$_V$1.2) channel activation (*Jaggar et al., 2000*). The ensuing increase in intracellular Ca$^{2+}$ concentration would stimulate vasoconstriction and an elevation in blood pressure, as observed here. PKD2 channel knockout abolishes the stimulus-induced inward Na$^+$ current, attenuating depolarization and vasoconstriction.

The splanchnic and skeletal muscle circulations account for up to 35 and 80 (during physical exertion) % of cardiac output, respectively. Changes in arterial contractility within these organs leads to significant modification of total peripheral resistance and systemic blood pressure. We demonstrate that myocyte PKD2 channels regulate contractility in both of these physiologically important circulations. Our data also show that distinct mechanical and chemical stimuli activate PKD2 channels in these arteries to produce vasoconstriction. Intravasulcar pressure stimulates vasoconstriction in resistance-size arteries from a wide variety of tissues. Mice lacking $\alpha_1$-adrenergic receptors are hypotensive and have reduced vasoconstrictor responses to phenylephrine in mesenteric arteries, highlighting the relevance of this pathway to blood pressure regulation (*Tanoue et al., 2002*). $\alpha_1$-adrenergic receptor activation is reported to stimulate vasoconstriction through the release of intracellular Ca$^{2+}$ stores and the activation of voltage-dependent Ca$^{2+}$ channels and non-selective cation channels (*Utz et al., 1999*; *Inoue and Kuriyama, 1993*; *Eckert et al., 2000*). We show that intravascular pressure stimulates PKD2 channels in myocytes of hindlimb arteries, whereas phenylephrine activates PKD2 channels in mesenteric artery myocytes. Stimulus-induced PKD2 channel activation in these arteries leads to vasoconstriction. Surprisingly, we found that myocyte PKD2 channels do not contribute to the myogenic response in mesenteric arteries or to phenylephrine-induced vasoconstriction in hindlimb arteries. Angiotensin II-induced vasoconstriction did not require myocyte PKD2 channels in either hindlimb or mesenteric arteries. These data indicate that stimuli which activate myocyte PKD2 channels are vascular bed-specific, strengthening the concept that the regulation and function of a protein is not homogeneous in the circulation and findings in myocytes of one arterial bed should not simply be translated to other vasculature.

Arteries of diverse organs are exposed to unique environments, including the range of intravascular pressures and the types and concentrations of vasoconstrictor and vasodilator stimuli that regulate contractility. Thus, it would be surprising if the molecular components of intracellular signaling pathways were identical in arterial myocytes of different organs. It was outside of the central aim of this study and beyond scope to determine the differential mechanisms by which pressure and adrenergic receptors activate PKD2 channels in arteries of different organs. Several possibilities exist. Pressure-induced vasoconstriction was described over 100 years ago and yet the mechanosensor, intracellular signal transduction pathways and the ion channels that produce this physiological response are still unresolved (*Bayliss, 1902*). Candidates for the pressure mechanosensor include, one or more proteins that have been proposed in smooth muscle or non-smooth muscle cell types, proteins that have not yet been discovered, elements of the cytoskeleton, or ion channels that act alone, in series or in parallel with other proteins and may be homomultimeric or heteromultimeric proteins. Piezo1, angiotensin II type one receptors and GPR68 have been proposed to act as vascular mechanosensors (*Li et al., 2014*; *Xu et al., 2018*; *Schleifenbaum et al., 2014*; *Blodow et al., 2014*). Importantly, we show here that there is not a singular, homogeneous mechanism that mediates the myogenic response in all artery types, adding additional complexity to this question. It is possible that intravascular pressure mechanosensors are different in hindlimb and mesenteric artery myocytes. If this is the case, the mechanosensor protein(s) present in hindlimb artery myocytes activates PKD2 channels, whereas the mechanosensor(s) in mesenteric artery myocytes does not stimulate PKD2 and vice versa. The intracellular signaling pathways by which pressure and $\alpha_1$-adrenergic receptors activate PKD2 channels may also not be the same or may couple differently in myocytes of each artery type. Intracellular pathways that transduce the pressure signal to activate ion channels are not resolved, although $G_{q11}$-coupled phospholipase C (PLC) activation is one candidate (*Kauffenstein et al., 2012*). $\alpha_1$-adrenergic receptors also activate $G_{q/11}$. Thus, if pressure and $\alpha_1$-adrenergic receptors both stimulate $G_{q11}$, local or global proximity signaling to PKD2 channels may underlie differential signaling in each artery type. When considering the molecular identity of the channels that are involved, PKD2 can form both homotetramers and heteromultimers with other channels, including TRPC1 and TRPV4 (*Hanaoka et al., 2000*; *Köttgen et al., 2008*; *Zhang et al., 2009*; *Bai et al., 2008*; *González-Perrett et al., 2001*). Whether PKD2 forms homomers or heteromers with other TRP channels in arterial myocytes is unclear, but differential PKD2 channel tetramerization may also underlie the distinct activation of these proteins by pressure and $\alpha_1$-adrenergic receptors in hindlimb and mesenteric arteries. Other channels, including TRPC6, TRPM4 and ANO1 contribute to the myogenic tone in cerebral arteries, but their relationship to PKD2-mediated responses in mesenteric and hindlimb arteries is unclear (*Welsh et al., 2002*; *Earley et al., 2004*; *Bulley et al., 2012*). Conceivably, other ion channels may signal in series to PKD2, with the proteins involved and the sequence of events differing in mesenteric and hindlimb artery myocytes. Given the large number of unresolved signaling elements and the possible permutations that would require study, it was beyond scope to determine the differential mechanisms by which pressure and adrenergic receptors activate PKD2 channels in arteries of different organs. Future studies should investigate these possibilities.

Having established that myocyte PKD2 channels regulate arterial contractility and systemic blood pressure, we investigated whether hypertension is associated with an alteration in PKD2 expression, surface protein and function. Total and surface PKD2 protein were both higher in mesenteric and hindlimb arteries of hypertensive mice than in normotensive controls, indicating upregulation. PKD2 protein was primarily located in the plasma membrane in arteries of normotensive and hypertensive mice demonstrating that the distribution of channels was unchanged. These data indicate that hypertension is associated with an increase in the amount of PKD2 protein, which traffics to the plasma membrane, thereby increasing surface channel number. Myocyte-specific PKD2 knockout reduced both phenylephrine-induced vasoconstriction and systemic blood pressure and prevented an increase in wall-to-lumen thickness in hypertensive mice. The angiotensin II-induced elevation in blood pressure was smaller in *Pkd2* smKO mice than in *Pkd2*$^{fl/fl}$ controls, supporting the concept that a higher abundance of surface PKD channels in arterial myocytes was associated with vasoconstriction. Our data demonstrate that myocyte PKD2 channels are upregulated during hypertension and that PKD2 targeting reduces vasoconstriction, blood pressure and arterial remodeling during hypertension, eliciting multi-modal benefits.

When considering our findings, a discussion of human diseases that are known to be associated with PKD2 is warranted. Autosomal Dominant Polycystic Kidney Disease (ADPKD) is the most prevalent monogenic human disease worldwide, affecting 1 in 1000 people. ADPKD occurs due to mutations in either PKD1 or PKD2 proteins (*Torres et al., 2007*). Currently, more than 275 human variants in PKD2 have been identified (Autosomal Dominant Polycystic Kidney Disease Mutation Database, Mayo Clinic; http://pkdb.pkdcure.org). ADPKD is characterized by the appearance of renal cysts, although a significant proportion of patients with apparently normal renal function develop hypertension prior to the development of cysts (*Torres et al., 2007*; *Valero et al., 1999*; *Martinez-Vea et al., 2004*). Here, we show that myocyte PKD2 knockout reduced systemic blood pressure, which is in apparent contrast to the blood pressure increase observed in ADPKD patients. There are several explanations for these differential findings. First, the effects of complete PKD2 abrogation, such as in the knockout studied here, and that of a PKD2 mutation found in an ADPKD patient, on arterial myocyte function may differ. Second, a global PKD2 mutation in an ADPKD patient will alter the function of all cell types in which PKD2 is expressed. Such widespread changes in the physiology of many different cell types could alter arterial myocyte contractility and blood pressure through a variety of mechanisms, leading to net vasoconstriction even if myocyte PKD2 function is also compromised. Third, in ADPKD patients PKD2 mutation is constitutive, whereas here we studied the effects of PKD2 knockout over a three week period in fully developed adult mice. A constitutive PKD2 mutation may modify vascular development and may have chronic effects on myocyte signaling that would not occur in our study. Fourth, the loss of PKD2 function in arterial myocytes of ADPKD patients may attenuate the blood pressure elevation that occurs due to the loss of PKD2 function in other cell types. Future studies should be designed to investigate the effects of PKD2 mutations that occur in ADPKD patients on vascular myocyte function and systemic blood pressure. Our demonstration that myocyte PKD2 channels regulate blood pressure is a step forward to better understanding the significance of this myocyte ion channel in cardiovascular physiology and disease.

In summary, we show that arterial myocyte PKD2 channels are activated by distinct stimuli in arteries of different tissues, increase systemic blood pressure, are upregulated during hypertension and genetic knockout in vivo leads to vasodilation and a reduction in both physiological and high blood pressure.

## Materials and methods

**Key resources table**

| Reagent type (species) or resource | Designation | Source or reference | Identifiers | Additional information |
|---|---|---|---|---|
| Strain, strain background (*Mus musculus*) | Pkd2<sup>fl/fl</sup> | John Hopkins PKD Core | PMID:20862291 | Mice with *Pkd2* gene flanked by loxP regions |
| Strain, strain background (*M. musculus*) | *SMMHC-CreER<sup>T2</sup>Myh11-cre/ERT2* | Jackson Laboratories | Stock # 019079 PMID:18084302 | Mice with tamoxifen-inducible Cre recombinase that is expressed under the smooth muscle myosin heavy polipeptide 11 (*Myh11*) promoter. |
| Strain, strain background (*M. musculus*) | Pkd2<sup>fl/fl</sup>:smCre<sup>+</sup> | This paper | | Mouse line created in-house by mating Pkd2<sup>fl/fl</sup> with SMMHC-CreER<sup>T2</sup>. Mice with inducible smooth muscle-specific deletion of PKD2. |
| Antibody | anti PKD2 (rabbit polyclonal) | Baltimore PKD Core | Rabbit mAB 3374 CT-14/4 | IF 1:200 dilution |
| Antibody | anti PKD2 (mouse monoclonal) | Santa Cruz | Cat# sc-47734 RRID:AB_672380 | WB 1:100 dilution, IF 1:100 dilution |
| Antibody | anti PKD2 (mouse monoclonal) | Santa Cruz | Cat# sc-28331 RRID:AB_672377 | WB 1:100 dilution, IF 1:100 dilution |

*Continued on next page*

*Continued*

| Reagent type (species) or resource | Designation | Source or reference | Identifiers | Additional information |
|---|---|---|---|---|
| Antibody | anti PKD1 (mouse monoclonal) | Santa Cruz | Cat# sc-130554 | WB 1:100 dilution |
| Antibody | anti $Ca_V1.2$ (mouse monoclonal) | UC Davis/NIH NeuroMab | Cat# 73–053 RRID:AB_10672290 | WB 1:100 dilution |
| Antibody | anti ANO1 (rabbit monoclonal) | Abcam | Cat# ab64085 | WB 1:100 dilution |
| Antibody | anti TRPC6 (rabbit polyclonal) | Abcam | Cat# ab62461 | WB 1:1000 dilution |
| Antibody | anti TRPM4 (rabbit polyclonal) | Abcam | Cat# ab106200 | WB 1:500 dilution |
| Antibody | anti Actin (mouse monoclonal) | Millipore Sigma | Cat# MAB1501 | WB 1:5000 dilution |
| Antibody | Alexa 555 secondary antibodies (anti rabbit and anti mouse) | Thermofisher | Cat# A-21429 and # A-31570 | IF 1:400 dilution |
| Other | Nuclear staining (DAPI) | Thermofisher | Cat# 3571 | IF 1:1000 dilution |
| Commercial assay or kit | EZ-Link Sulfo-NHS -LC-LC-Biotin | Thermofisher | Cat# 21338 | |
| Commercial assay or kit | EZ-Link Maleimide-PEG2-Biotin | Thermofisher | Cat# 21901BID | |
| Commercial assay or kit | Mouse Angiotensin II ELISA kit | Elabscience | Cat# E-EL-M2612 | |
| Commercial assay or kit | Mouse Atrial Natriuretic Peptide ELISA kit | Elabscience | Cat# E-EL-M0166 | |
| Commercial assay or kit | Mouse Aldosterone ELISA kit | Mybiosource | Cat# MBS775626 | |
| Chemical compound, drug | Angiotensin II | Sigma-Aldrich | Cat# A9525 | |

### Animals

All procedures were approved by the Animal Care and Use Committee of the University of Tennessee. *Pkd2^{fl/fl}* mice with *loxP* sites flanking exons 11–13 of the *Pkd2* gene were obtained from the John Hopkins PKD Core. *Pkd2^{fl/fl}* mice were crossed with tamoxifen-inducible smooth muscle-specific Cre mice (*Myh11-cre/ERT2*, Jackson Labs, *Zhang et al., 2009*) to produce *Pkd2^{fl/fl}:Myh11-cre/ERT2* mice. Male *Pkd2^{fl/fl}:Myh11-cre/ERT2* or *Pkd2^{fl/fl}* mice (6–10 weeks of age) were injected with tamoxifen (1 mg/ml, i.p.) once per day for 5 days and studied no earlier than 17 days after the last injection. C57BL/6J mice (12 weeks old) were purchased from Jackson Laboratories. Angiotensin II (1.5 ng/g/min) and saline (0.9 NaCl) were infused in mice using subcutaneous osmotic minipumps (Alzet).

### Tissue preparation and myocyte isolation

Mice were euthanized with isofluorane (1.5%) followed by decapitation. First- to fifth-order mesenteric and hindlimb (saphenous, popilital and gastrocnemius) arteries were removed and placed into ice-cold physiological saline solution (PSS) that contained (in mM): 112 NaCl, 6 KCl, 24 $NaHCO_3$, 1.8 $CaCl_2$, 1.2 $MgSO_4$, 1.2 $KH_2PO_4$ and 10 glucose, gassed with 21% $O_2$, 5% $CO_2$ and 74% $N_2$ to pH 7.4. Arteries were cleaned of adventitial tissue and myocytes dissociated in isolation solution containing (in mM): 55 NaCl, 80 sodium glutamate, 5.6 KCl, 2 $MgCl_2$, 10 HEPES and 10 glucose (pH 7.4, NaOH) using enzymes, as previously described (*Qian et al., 1997*).

### Genomic PCR

Genomic DNA was isolated from mesenteric and hindlimb arteries using a Purelink Genomic DNA kit (ThermoFisher Scientific). Reaction conditions used are outlined in the Baltimore PKD core center genotyping protocol (http://baltimorepkdcenter.org/mouse/PCR%20Protocol%20for%

20Genotyping%20PKD2KO%20and%20PKD2%5Eneo.pdf). Genotyping was performed using a 3-primer strategy, with primers a (5'-CCTTTCCTCTGGTTCTGGGGAG), b (5'-GTTGATGCTTAGCAGATGATGGC) and c (5'-CTGACAGGCACCTACAGAACAGTG) used to identify floxed and deleted alleles.

## RT-PCR

Fresh, dissociated mesenteric artery myocytes were manually collected using an enlarged patch pipette under a microscope. Total RNA was extracted from ~ 500 myocytes using the Absolutely RNA Nanoprep kit (Agilent Technologies, Santa Clara, CA, USA). First-strand cDNA was synthesized from 1 to 5 ng RNA using SuperScript IV (Invitrogen, Life Technologies). PCR was performed on first-strand cDNA using the following conditions: an initial denaturation at 94°C for 2 min, followed by 40 cycles of denaturation at 94°C for 30 s, annealing at 56°C for 30 s, and extension at 72°C for 1 min. PCR products were separated on 2% agarose–TEA gels. Primers were used to amplify transcripts of PKD2, aquaporin 4, myosin heavy chain 11, platelet-endothelial cell adhesion molecule 1 (PECAM-1) and actin (*Supplementary file 1*). The PKD2 forward primer spanned the junction of exons 9 and 10 and the reverse primer annealed to exon 13.

## Western blotting

Proteins were separated on 7.5% SDS-polyacrylamide gels and blotted onto PVDF membranes. Membranes were blocked with 5% milk and incubated with the following primary antibodies: $Ca_V1.2$ (Neuromab), PKD1 and PKD2 (Santa Cruz, sc-100415), ANO1, TRPC6 and TRPM4 (Abcam) and actin (MilliporeSigma) overnight at 4°C. Membranes were washed and incubated with horseradish peroxidase-conjugated secondary antibodies at room temperature. Protein bands were imaged using an Amersham Imager 600 gel imaging system (GE Healthcare) and quantified using ImageJ software.

## *En-face* arterial immunofluorescence

Arteries were cut longitudinally and fixed with 4% paraformaldehyde in PBS for 1 hr. Following a wash in PBS, arteries were permeabilized with 0.2% Triton X-100, blocked with 5% goat serum and incubated overnight with PKD2 primary antibody (Rabbit mAB 3374 CT-14/4: Baltimore PKD Center) at 4°C. Arteries were then incubated with Alexa Fluor 555 rabbit anti-mouse secondary antibody (1:400; Molecular Probes) and 4',6-diamidino-2-phenylindole, dihydrochloride (DAPI) (1:1000; Thermo Scientific) for 1 hr at room temperature. Segments were washed with PBS, oriented on slides with the endothelial layer downwards and mounted in 80% glycerol solution. DAPI and Alexa 555 were excited at 350 nm and 555 nm with emission collected at $\leq$437 nm and $\geq$555 nm, respectively.

## Isolated arterial myocyte immunofluorescence

Myocytes were plated onto poly-L-lysine-coated coverslips, fixed with 3.7% paraformaldehyde in PBS and permeabilized with 0.1% Triton X-100. After blocking with 5% BSA, cells were incubated with mouse monoclonal anti-PKD2 antibody (Santa Cruz) overnight at 4°C. Slides were then washed and incubated with Alexa Fluor 555 rabbit anti-mouse secondary antibody (Molecular Probes). Secondary antibodies were washed and coverslips mounted onto slides. Images were acquired using a Zeiss 710 (Carl Zeiss) laser-scanning confocal microscope and 40x and 63x oil immersion objectives.

## Kidney histology

Kidney sections were stained with H and E and examined by Probetex, Inc (San Antonio, Texas). Briefly, tubules, glomeruli and vasculature were examined for frequency or homogeneity of pathologic abnormalities. These included characteristics of hypercellularity, hypocellularity, necrosis, apoptosis, matrix accumulation, inflammation, fibrosis and protein droplets. The size and thickness of the cortex, medulla, papilla, glomeruli, tubules and vasculature were also examined.

## Telemetric blood pressure measurements

Radiotelemetric transmitters (PA-C10, Data Sciences International) were implanted subcutaneously into anesthetized mice, with the sensing electrode placed in the aorta via the left carotid artery. Seven days later blood pressures were measured using a PhysioTel Digital telemetry platform (Data Sciences International). Dataquest A.R.T. software was used to acquire and analyze data.

### Echocardiography

Mice were anesthetized with isofluorane. Ultrasound gel was placed on a hairless area of the chest before and echocardiography performed using a Visual Sonics Vevo 2100 system. Anesthetic depth, heart rate and body temperature were monitored throughout the procedure.

### Arterial histology

Arteries were fixed with paraformaldehyde and embedded in paraffin. 5 μm thick sections were cut using a microtome and mounted on slides. Sections were de-paraffinized, blocked in BSA and incubated with H and E. Images were acquired using a transmitted light microscope (Nikon Optiphot-2) and measurements made using Stereo Investigator software (MicrobrightField, Inc). Wall-to-lumen ratios were calculated as wall thickness/lumen diameter, where the wall (tunica media) thickness and lumen diameter of each section was the averages of four and two equidistant measurements, respectively.

### Blood and urine analysis

Retro-orbital blood was drawn from isofluorane-anesthetized mice using a Microvette Capilliary Blood Collection System (Kent Scientific Corporation). Plasma was extracted and angiotensin II (Elabscience), aldosterone (Mybiosource) and atrial natriuretic peptide (Elabscience) concentrations measured using commercially available ELISA kits and an EL800 plate reader (BioTeK). Mice were housed in individual metabolic cages for 72 hr and urine collected for the final 24 hr. Plasma and urine electrolyte concentrations were measured using the MMPC Core at Yale University.

### Pressurized artery myography

Experiments were performed using isolated mouse third-, fourth and fifth-order mesenteric arteries and first-order gastrocnemius arteries using PSS gassed with 21% $O_2$/5% $CO_2$/74% $N_2$ (pH 7.4). Arterial segments 1–2 mm in length were cannulated at each end in a perfusion chamber (Living Systems Instrumentation) continuously perfused with PSS and maintained at 37°C. Intravascular pressure was altered using an attached reservoir and monitored using a pressure transducer. Luminal flow was absent during experiments. Arterial diameter was measured at 1 Hz using a CCD camera attached to a Nikon TS100-F microscope and the automatic edge-detection function of IonWizard software (Ionoptix). Myogenic tone was calculated as: 100 x (1-$D_{active}$/$D_{passive}$) where $D_{active}$ is active arterial diameter, and $D_{passive}$ is the diameter determined in the presence of $Ca^{2+}$-free PSS supplemented with 5 mM EGTA).

### Perfused hindlimb pressure measurements

Isolated hindlimbs were inserted into a chamber containing gassed PSS (21% $O_2$/5% $CO_2$/74% $N_2$) that was placed on a heating pad to maintain temperature at 37°C. The femoral artery was cannulated with a similar diameter glass pipette and perfused with gassed PSS at 37°C using a peristaltic pump. Perfusion pressure was measured using a pressure transducer connected to the inflow. The flow rate was increased stepwise from 0 to 2.5 mL/min in 0.25 mL/min steps to generate a response curve. Values were corrected by subtracting the pressure produced by the pipette alone at each flow rate. Prior to measuring responses to phenylephrine, flow rate was adjusted to maintain a constant perfusion pressure of 80 mmHg. Data were recorded and analyzed using IonWizard software (Ionoptix).

### Wire myography

Mesenteric artery segments (1 st and 2nd order, 2 mm in length) were mounted on tungsten wires in a multi-channel myography system (Danish Myo Technology). Chambers contained PSS that was continuously gassed with 21% $O_2$/5% $CO_2$/74% $N_2$ (pH 7.4). Arterial rings were subjected to a resting tension of 10 mN and allowed to equilibrate prior to experimentation. Responses were measured to increasing concentrations of phenylephrine or 60 mM $K^+$. Data were acquired and analyzed using LabChart software (ADInstruments).

## Pressurized artery membrane potential measurements

Membrane potential was measured by inserting sharp glass microelectrodes (50–90 m$\Omega$) filled with 3 M KCl into the adventitial side of pressurized mesenteric or hindlimb arteries. Membrane potential was recorded using a WPI FD223a amplifier and digitized using a MiniDigi 1A USB interface, pClamp 9.2 software (Axon Instruments) and a personal computer. Criteria for successful intracellular impalements were: (1) a sharp negative deflection in potential on insertion; (2) stable voltage for at least 1 min after entry; (3) a sharp positive voltage deflection on exit from the recorded cell and (4) a < 10% change in tip resistance after the impalement.

## Patch-clamp electrophysiology

Isolated arterial myocytes were allowed to adhere to a glass coverslip in a recording chamber. The conventional whole-cell configuration was used to measure non-selective cation currents ($I_{cat}$) by applying voltage ramps (0.13 mV/ms) between −100 mV and +100 mV from a holding potential of −40 mV. For cell swelling experiments, the pipette solution contained (in mM): Na$^+$ aspartate 115, mannitol 50, HEPES 10, glucose 10, EGTA 1, NaGTP 0.2, with free Mg$^{2+}$ and Ca$^{2+}$ of 1 mM and 200 nM, respectively (pH 7.2, NaOH). Isotonic (300 mOsm) bath solution contained (in mM): Na$^+$ aspartate 115, mannitol 50, glucose 10, HEPES 10, CaCl$_2$ 2, MgCl$_2$ 1 (pH 7.4, NaOH). Hypotonic (250 mOsm) bath solution was the same formulation as isotonic bath solution with the exclusion of mannitol (pH 7.4, NaOH). For experiments that measured $I_{Cat}$ regulation by phenylephrine, the bath solution contained (in mM): 140 NaCl, 10 glucose, 10 HEPES, 1 MgCl$_2$, and pH was adjusted to 7.4 with NaOH. Pipette solution contained: 140 NaCl, 10 HEPES, 10 Glucose, 5 EGTA, 1 MgATP, 0.2 NaGTP, and pH was adjusted to 7.2 with NaOH. Total MgCl$_2$ and total CaCl$_2$ were adjusted to give free Mg$^{2+}$ concentrations of 1 mM and free Ca$^{2+}$ of 200 nM. Free Mg$^{2+}$ and Ca$^{2+}$ were calculated using WebmaxC Standard (http://www.stanford.edu/~cpatton/webmaxcS.htm). Na$^+$ concentration was reduced in bath solutions through equimolar replacement with N-methyl-D-glucamine. Liquid junction potentials that occurred due to a reduction in bath Na$^+$ concentration were measured experimentally and used to correct reversal potential values measured in myocytes during voltage-clamp experiments. Briefly, liquid junction potentials were determined by measuring the immediate shift in voltage that occurred in a patch-clamp pipette containing pipette solution when bath solution containing normal Na$^+$ concentration was replaced with one containing low Na$^+$. Currents were recorded using an Axopatch 200B amplifier and Clampex 10.4 (Molecular Devices), digitized at 5 kHz and filtered at 1 kHz. Offline analysis was performed using Clampfit 10.4.

## Arterial biotinylation

Procedures used were similar to those previously described (*Tsiokas et al., 1997*). Briefly, arteries were biotinylated with EZ-Link Sulfo-NHS-LC-LC-Biotin and EZ-Link Maleimide-PEG2-Biotin. Unbound biotin was quenched with glycine-PBS, washed with PBS and then homogenized in radio-immunoprecipitation assay (RIPA) buffer. Protein concentration was normalized and biotinylated surface protein was captured by incubating cell lysates with avidin beads (Pierce) at 4°C. Proteins present in biotinylated and non-biotinylated samples were identified using Western blotting.

## Statistical analysis

OriginLab and GraphPad InStat software were used for statistical analyses. Values are expressed as mean ± SEM. Student t-test was used for comparing paired and unpaired data from two populations and ANOVA with Student-Newman-Keuls post hoc test used for multiple group comparisons. p<0.05 was considered significant. Power analysis was performed to verify that the sample size gave a value of > 0.8 if $P$ was > 0.05.

# Acknowledgements

This study was supported by NIH/NHLBI grants HL67061, HL133256 and HL137745 to JHJ, American Heart Association (AHA) Scientist Development Grants to SB and MDM and AHA postdoctoral fellowship to RH. Studies utilized resources provided by the NIDDK-sponsored Johns Hopkins Polycystic Kidney Disease Research and Clinical Core Center, P30 DK090868.

## Additional information

### Funding

| Funder | Grant reference number | Author |
|---|---|---|
| American Heart Association | 16SDG27460007 | Simon Bulley |
| American Heart Association | 16POST30960010 | Raquibul Hasan |
| American Heart Association | 15SDG22680019 | M Dennis Leo |
| National Heart, Lung, and Blood Institute | HL67061 | Jonathan H Jaggar |
| National Heart, Lung, and Blood Institute | HL133256 | Jonathan H Jaggar |
| National Heart, Lung, and Blood Institute | HL137745 | Jonathan H Jaggar |

The funders had no role in study design, data collection and interpretation, or the decision to submit the work for publication.

### Author contributions

Simon Bulley, Conceptualization, Investigation, Visualization, Methodology, Writing—review and editing; Carlos Fernández-Peña, Formal analysis, Investigation, Visualization, Writing—review and editing; Raquibul Hasan, M Dennis Leo, Padmapriya Muralidharan, Formal analysis, Investigation; Charles E Mackay, Kirk W Evanson, Luiz Moreira-Junior, Alejandro Mata-Daboin, Sarah K Burris, Qian Wang, Korah P Kuruvilla, Investigation; Jonathan H Jaggar, Conceptualization, Funding acquisition, Visualization, Methodology, Writing—original draft, Writing—review and editing

### Author ORCIDs

Simon Bulley (iD) http://orcid.org/0000-0001-5985-0489
Carlos Fernández-Peña (iD) http://orcid.org/0000-0002-0726-3204
Jonathan H Jaggar (iD) http://orcid.org/0000-0003-1505-3335

### Ethics

Animal experimentation: This study was performed in strict accordance with the recommendations in the Guide for the Care and Use of Laboratory Animals of the National Institutes of Health. All of the animals were handled according to an approved institutional animal care and use committee (IACUC) protocol (#17-068.0) of the University of Tennessee. Every effort was made to minimize suffering.

### Decision letter and Author response

Decision letter https://doi.org/10.7554/eLife.42628.sa1
Author response https://doi.org/10.7554/eLife.42628.sa2

## Additional files

### Supplementary files

• Supplementary file 1. Primers used for RT PCR. The PKD2 forward primer recognizes nucleotides in exon 9 and 10 and the reverse primer was aligned with a sequence in exon 13.

• Transparent reporting form

### Data availability

All data generated or analysed during this study are included in the manuscript and supporting files.

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
