## [Decision Letter]

[Editors’ note: a previous version of this study was rejected after peer review, but the authors submitted for reconsideration. The first decision letter after peer review is shown below.]

Thank you for submitting your work entitled "Arterial smooth muscle cell PKD2 (TRPP1) channels control systemic blood pressure" for consideration by *eLife*. Your article has been reviewed by three reviewers and the evaluation has been overseen by Kenton Swartz as the Reviewing Editor and a Senior Editor. The following individuals involved in review of your submission have agreed to reveal their identity: Alexander Chesler (Reviewer #1).

Our decision has been reached after consultation between the reviewers. Based on these discussions and the individual reviews below, we regret to inform you that your work will not be considered further for publication in *eLife*.

Although all three reviewers were somewhat divergent in their overall enthusiasm for your work, after discussion the consensus was that the characterization of the PKD2 channel in the various cell types examined was incomplete and would require considerable additional experiments to provide compelling support for the core conclusions. In particular, all three reviewers felt that more experiments would be required to better explain the different results obtained in mesenteric and hind limb smooth muscle.*eLife* We would be willing to consider a new manuscript if you decide to undertake the suggested experiments and the results help to resolve the issues raised by the reviewers.

*Reviewer #1:*

The manuscript by Bulley and colleagues addresses the molecular mechanisms by systemic blood pressure is regulated. Specifically, they look at the role of the TRP channel PKD2 by examining mice where the channel is knocked out selectively from smooth muscle cells in adults.

As they correctly point out, there is much contradictory data about how pressure is sensed by cells in vasculature and the function of TRP channels. Here they chose to focus on PKD2 since previous studies established that it is expressed in myocytes, has been previously proposed to be a mechanosensor, and perturbing its function negatively affects myogenic tone.

Overall this is very straight solid and systematic study from a lab with a track record of probing the physiology of the vascular system. They make a convincing case that PKD2 is important for regulating blood pressure. As such, I have relatively minor comments:

1) The word "control" in the Title is far too strong an interpretation of their findings. While I agree that they help clarify PKD2's importance, the effects, although significant, highlight there clearly exist other equally important factors/mechanisms.

2) They place too much emphasis/value on the generation of mouse cross and the interpretation of their findings. While they are correct using an inducible strategy is a useful, and probably necessary approach, it is also fairly standard these days. Words like first and novel are not really appropriate and are distracting. Genotyping information does not require a figure and a fair amount of the descriptions should go into the Materials and methods section.

3) The most confusing part of the study is the differential effects of PKD2 deletion on hindlimb as compared to mesenteric arteries. This needs to be discussed and explained in greater detail.

4) How is PKD2 seemingly sensing pressure in one cell but not the other? Osmolarity is a poor proxy for pressure. Recordings using direct mechanical stimulation (Piezo-driven glass probes and/or pressure clamp recordings) should be done for both myocyte types.

5) Along this same line, are the same types of adrenergic receptors expressed in both types of myocytes? How do they explain the differential effects of PKD2 deletion on phenylephrine responses?

6) Furthermore, given they see differential effects between myocyte, where else is the SMMHC-CreERT2 expressed? Is Tamoxifen treatment equally efficient for knocking out PKD2 in all places?

7) It has become standard to show the individual data points for each bar graph so the reader can more fully appreciate the sample size, distribution and effect sizes.

8) Although they discuss TRP channels, they should also consider the roles of Piezo1 and Piezo2. Are these expressed in myocytes? Are they dysregulated after PKD2 deletion?

9) Similarly, GPCRs such as the angiotensin receptor and (very recently) GPR68 have been implicated in mechanosensing. Are these found in myocytes? Could PKD2 be functioning downstream? Some more discussion of what they think PKD2 is doing would be helpful.

10) The summary bar graphs as provided are insufficient to evaluate the sample size, distribution and effect sizes.

*Reviewer #2:*

This descriptive manuscript examines impact of smooth muscle-specific, genetic attenuation of the PKD2 gene- which encodes for the polcystin-2 ion channel- in the vascular system. Prior knowledge was limited to indiscriminate knockout of the gene in mouse models and generally focused on the renal system, where variants in PKD2 are associated with kidney cyst formation in autosomal dominant polycystic kidney disease (ADPKD). The authors use invitro and in vivo techniques to explore the mechanism of this TRP channel's putative regulation of vascular tone in response to osmotic stretch and adrenergic stimuli. Using their mouse model, among other observations they observe that conditional PKD2 knock out animals are hypotensive but they also observe seemingly conflicting results regarding how the Polycystin-2 ion channel causes this phenotype. Although not stated, there appears to be no difference in the Polycystin-2 channels constitutive activity, but the channel activity is revealed using two different stimuli that is location dependent. On one hand, PKD2/polycystin-2 in the appears to be an osmosensitive cation channel in the hindlimb artery. On the other hand, the same channel is not osmosensitive but is activated by adrenergic stimuli in the mesenteric artery. But it's difficult to see how this basic TRP gating mechanism is lost from one location to the next, and together with the incomplete electrophysiology analysis, makes me wonder if the currents they attribute to PKD2 are the same channel in both places? Thus, the conclusions need to be strengthened with the following experiments.

1) Determine if a PKD2-dependent single channel conductance can be measured from the plasma membrane and compare them from the mesenteric and hindlimb smooth muscle cells.

2) Determine if the putative PKD2 currents are cation-selective from the mesenteric and hindlimb smooth muscle. The authors used symmetric NaCl or NaAsp solutions and conclude that this channel is Na-permeable or is "cation-selective" without demonstrating this, as either anion can permeate chloride channels in smooth muscle (PMID: 10087337). This is particularly important for the osmo-regulation experiments, as the 300-250mOs condition invariably causes an increasing in the ohmic leak currents and likely activates resident swell-activated Cl^-^ channels.

3) The reversal potentials and voltage-dependence are variable between the electrophysiology recordings that report the putative PKD2 channel activity. Thus, uniform steady state analysis of the current should be per An example of this is Figure 4 —figure supplement 1and Figure 6—figure supplement 1. In Figure 6—figure supplement 1the reversal potential shifts from approximately -10 mV to +10 mV between the knockout and control currents (respectively), but it is zero millivolts experiments under the same conditions. Perhaps there are differences in the relative ion permeability of these cell membranes, that is variable from cell to cell, but cannot be considered a pure PKD2 current.

Whether hypertension in ADPKD is a primary vasculopathy secondary to mutations in the polycystin genes or perhaps, secondary to activation of the renin-angiotensin-aldosterone system by cyst expansion and intrarenal ischemia is unclear and thus is an intriguing question. However, it is important to point out that hypertension, not hypotension (as observed in their mouse) is associated with ADPKD in man. Thus, while the findings are descriptive of PKD2 regulation of vascular tone, it does little to enhance our understanding the disease complications associated with this channel.

The results from the bar graphs should be enumerated on an excel file.

*Reviewer #3:*

The mouse model is interesting and experiments employing it provide compelling evidence to support a role for smooth muscle PKD2 in blood pressure regulation. This facet of the current manuscript is novel and by far the best developed.

On the other hand, findings related to differential stimulus-sensitivity of PKD2 in hindlimb arteries (pressure-induced tone) and mesenteric arteries (PE-induced construction) are puzzling and leaves one hanging. There is an unsatisfying lack of a physiological rationale for this outcome. Which one is related to blood pressure regulation, or are both, and which one is more important? How would you prove this?

Having said that, the effects of smooth muscle PKD2 on blood pressure regulation are striking and convincing. The effects of smooth muscle deletion of PKD2 on pressure and PE-induced increases on blood vessel diameter are convincing. The effects on membrane current are incomplete and not convincing. In my view, the authors could turn this manuscript into a compelling study, without much experimentation, with a much more balanced and nuanced presentation. For example, stressing the importance and novelty of the role of smooth muscle PKD2 in blood pressure regulation, and using the effects on myogenic and PE-induced tone as possible connectors while discussing the uncertainties of the connections.

1) Introduction: The motivation and rationale for the study are not well-developed. For example, in the first paragraph, it is stated "However the molecular identity of arterial myocyte ion channels that regulate blood pressure, and their mechanisms of modulation in vivo (are) poorly understood." This is not entirely true. L-type voltage-dependent calcium channels in vascular smooth muscle regulate blood pressure and are the targets of a major class of anti-hypertensive drugs. Furthermore, the "in vivo" modulation of any type of ion channel is not completely understood. The current study only sheds light on PKD2 modulation in vitro, and not in vivo.

2) Introduction and elsewhere: It is stated a number of times that the PKD2 knockout is in "systemic artery myocytes". This not true; it is all smooth muscle (vascular and non-vascular). This should be made clear. Do the mice have any other defects?

3) Subsection “Swelling activates PKD2 channels in hindlimb artery myocytes”: Cell-swelling-based electrophysiology data are incomplete. The authors show that cell swelling stimulates a TRP-like current in hindlimb artery myocytes, consistent with the demonstration that PKD2 plays a role in a mechanosensitive response (pressure-induced constriction) in hindlimb arteries. Although they report that PKD2 is not involved in pressure-induced constriction in mesenteric arteries, whether swelling-induced TRP-like currents are absent or present in myocytes from these arteries is not tested.

4) It is unclear why isotonic Na^+^ was used. A physiological solution could have been used, so that swelling-activated (Na+) current over physiological membrane potentials can be examined. Furthermore, a similar protocol was used by Welsh et al., 2002, with the conclusion that the cation current was through TRPC6 channels. This study should be discussed.

5) Results section: PE-based electrophysiology data are incomplete. The authors show that PE stimulates a TRP-like current in mesenteric artery myocytes, consistent with the demonstration that PE induces constriction of mesenteric arteries. The companion patch-clamp experiments (PE on hindlimb artery myocytes) are nowhere to be found. Since other TRP channels are present in these cells (at minimum, TRPM4, TRPC6 and TRPV4, according to their own presentation) and should be detectable under these conditions, this result is puzzling.

6) Not at all clear why PKD2 KO should have a dramatic effect on myogenic tone in hindlimb (and cerebral arteries) arteries, but no effect at all in mesenteric arteries. Seemingly inexplicable results are often the takeoff point for interesting scientific stories, but they are intellectually unsatisfying if left hanging, as they are here. Again, at first blush, it makes no sense that PKD2 KO would have a dramatic effect on PE effects in mesenteric arteries, but no effect in hindlimb arteries.

[Editors’ note: what now follows is the decision letter after the authors submitted for further consideration.]

Thank you for submitting your article "Arterial smooth muscle cell PKD2 (TRPP1) channels regulate systemic blood pressure" for consideration by *eLife*. Your article has been reviewed by three peer reviewers, and the evaluation has been overseen by Kenton Swartz as the Reviewing Editor and Richard Aldrich as the Senior Editor. The reviewers have opted to remain anonymous.

The reviewers have discussed the reviews with one another and the Reviewing Editor has drafted this decision to help you prepare a revised submission.

Summary:

This comprehensive study provides compelling evidence that arterial smooth muscle PKD2 (TRPP1) channels regulate blood pressure through transducing intravascular pressure (hindlimb) or sympathetic simulation into vasoconstriction. This is accomplished using a new inducible, smooth muscle-specific knockout of PKD2. The mystery of differential sensitivity of PKD2 to different stimuli remains to be explored and will be interesting to unravel. All in all, the authors have done an admirable job of revising the manuscript in response to the initial review.

Essential revisions:

1) The representative traces in Figure 4C can be confusing when compared to the Gd-sensitive currents plotted in Figure 4D. Please show traces for Gd for both conditions studied in Figure 4C and explicitly point out in the text what is shown in Figure 4C and D. Also, the graphs in Figures 4D and Figure 4—figure supplement 1B are confusing because only data for -100 and +100 are shown, yet they appear on a continuous x-axis labeled voltage. The x axis should be removed, and data plotted at bar graphs with conditions clearly labeled.

2) It appears that some of the changes made in revision were done hastily, as there are ambiguous statements and grammatical errors in the red text. For example, these lines from the methods are non-sequiturs "Equimolar N-methyl-glucamine (NMDG) was substituted for Na^+^. Liquid junction potential was measured experimentally." Substituted for which solution? What was the LJ potential then if it was measured and what was done with that measurement? Please take your time revising the text and figures to make sure everything will be clear to readers.

[Editors' note: further revisions were requested prior to acceptance, as described below.]

Thank you for resubmitting your work entitled "Arterial smooth muscle cell PKD2 (TRPP1) channels regulate systemic blood pressure" for further consideration at *eLife*. Your revised article has been favorably evaluated by Richard Aldrich (Senior Editor) and Kenton Swartz as the Reviewing Editor.

The manuscript has been nicely improved but there are some remaining issues that need to be addressed before acceptance, as outlined below:

Your revised manuscript has addressed our specific requests of the reviewing editor and reviewers. However, in evaluating the full manuscript it became evident that the conceptual problem you have nicely corrected in Figure 4E of the revised manuscript needs to also be applied to Figure 4—figure supplement 1, Figure 6D and Figure 6—figure supplement 1. We request that you make the appropriate changes to those remaining figures before publication.

---

## [Author Response]

[Editors’ note: the author responses to the first round of peer review follow.]

Reviewer #1:[…] 1) The word "control" in the Title is far too strong an interpretation of their findings. While I agree that they help clarify PKD2's importance, the effects, although significant, highlight there clearly exist other equally important factors/mechanisms.

As suggested, we have modified the Title.

2) They place too much emphasis/value on the generation of mouse cross and the interpretation of their findings. While they are correct using an inducible strategy is a useful, and probably necessary approach, it is also fairly standard these days. Words like first and novel are not really appropriate and are distracting. Genotyping information does not require a figure and a fair amount of the descriptions should go into the Materials and methods section.

We have removed all but one of these terms from the manuscript. We consider the mouse model generated here to be important step in the study of arterial smooth muscle TRP channels. Therefore, we retained one reference (in the abstract) to this paper being the first study to develop an inducible, cell-specific knockout of a TRP protein. As suggested, we have removed Supplemental figure 1, which was a schematic illustrating the generation of the mouse model, breeding strategy and the genotyping approach. More of this information is now discussed in the Materials and methods section.

3) The most confusing part of the study is the differential effects of PKD2 deletion on hindlimb as compared to mesenteric arteries. This needs to be discussed and explained in greater detail.

We have expanded the text in the sixth paragraph of the Discussion section to explain why PKD2 channels may be regulated by different stimuli in arterial myocytes of different organs. Arteries of diverse organs are exposed to unique environments, including range of intravascular pressures and the types and concentrations of vasoconstrictor and vasodilator stimuli that regulate contractility. It would be surprising if the molecular components of signaling pathways were identical in myocytes of different arteries, including those in the mesentery and hindlimb.

4) How is PKD2 seemingly sensing pressure in one cell but not the other? Osmolarity is a poor proxy for pressure. Recordings using direct mechanical stimulation (Piezo-driven glass probes and/or pressure clamp recordings) should be done for both myocyte types.

The goal of our study was to test the hypothesis that an arterial myocyte TRP channel regulates systemic blood pressure. We show that vasoconstrictor stimuli activate PKD2 channels in arterial myocytes, leading to Na^+^ influx, membrane depolarization, vasoconstriction and an elevation in blood pressure. It is not clear what the most appropriate stimulus is to apply to an isolated smooth muscle cell to simulate intravascular pressure in an artery where myocytes are positioned in cylindrical orientation. Cell swelling has been used as a standard method in many publications to stimulate mechanosensitive responses arterial myocytes. Swelling has been successfully demonstrated to activate currents, including non-selective cation and chloride currents, in myocytes. In these studies, the same channels activated by cell-swelling contributed to pressure-induced vasoconstriction, suggesting that osmolarity is a good proxy for pressure in isolated arterial myocytes. We used the same approach here and show that cell-swelling activates a Na^+^ current in hindlimb artery myocytes of PKD2^fl/fl^ mice but does not activate the same current in hindlimb artery myocytes of inducible, smooth muscle cell-specific PKD2 knockout mice. These results are supported by extensive data obtained using the best possible proxy for pressure, which is pressure itself, and show that pressure-induced depolarization and vasoconstriction in hindlimb arteries and perfusion pressure in intact hindlimb muscle are all attenuated in PKD2^sm-/-^ mice. Furthermore, we show using telemetry that PKD2 knockout reduces systemic blood pressure in conscious mice. We consider that this combination of approaches strongly supports our conclusion that pressure stimulates PKD2 channels in arterial myocytes.

It is unclear what is the best approach to use to simulate the force produced by intravascular pressure in an intact artery when studying an isolated arterial myocyte situated in a patch-clamp chamber. It is not known whether applying suction to cell-attached patches or pressing piezo-driven glass probes onto isolated myocytes are better proxys for intravascular pressure than cell swelling. To attempt to determine whether an alternative method to cell swelling also stimulates PKD2 channels in arterial myocytes, we performed patch-pressure experiments, similarly to what we did previously for TMEM16A channels in cerebral artery myocytes^1^. We also used glass probes. We found that applying patch pressure did activate single non-selective cation channels in hindlimb artery myocytes of PKD2^fl/fl^ mice, although this only occurred in a small percentage (3/30 = 10% ) of cell-attached patches. Given that the surface area of a cell-attached patch is ~1 µm^2^ and that of an arterial myocyte is ~3141 µm^2^ (assuming a 10 µm diameter cylinder that is 100 µm long), this low probability of obtaining channels within small membrane patches is not surprising. Because of the low probability of obtaining a patch containing channels, we would have to record from a very large number of patches (at least 100) to obtain sufficient data to analyze channel properties in PKD2^fl/fl^ myocytes. We would then be required to record from a similarly large number (again at least 100) of PKD2^sm-/-^ myocytes to determine whether the same pressure-activated single channel conductance is absent in the knockout, confirming that it is PKD2. We consider this amount of experimentation beyond the scope of the current study, particularly as we show that swelling-included Na^+^ currents, pressure-induced depolarization, pressure-induced vasoconstriction, muscle perfusion pressure, and systemic blood pressure are all attenuated in myocytes, arteries, whole hindlimb and in vivo circulation of PKD2^sm-/-^ mice. We also attempted to measure current regulation by piezo-driven glass probes, but encountered technical challenges related to the fact that we record from fresh-isolated, myocytes that do not strongly adhere to the glass coverslip in the patchclamp chamber. When we applied pressure to myocytes using a glass probe, the cells rolled, which is not unexpected given their cylindrical shape. This rolling will produce large variability in the amount of pressure applied to individual cells. While we consider that glass probes may be useful for future studies, there will be significant technical challenges that need to be overcome before such an approach can be used reproducibly for arterial myocytes. We consider that at present this is also beyond the scope of the current study. We trust that you consider the wide variety of cellular, whole artery, intact tissue and in vivo approaches we have used to support our conclusions that PKD2 channels contribute to pressure-induced vasoconstriction in hindlimb arteries. We also hope you recognize that major technical challenges exist with your request, making it beyond the scope of the current study.

5) Along this same line, are the same types of adrenergic receptors expressed in both types of myocytes? How do they explain the differential effects of PKD2 deletion on phenylephrine responses?

We have performed new experiments and now show that adrenergic receptor subtypes 1A, 1B and 1D are similar in mesenteric and hindlimb arteries of PKD2^fl/fl^ and PKD2^sm-/-^ mice (new Figure 1—figure supplement 3). These data indicate that the differential effects of PKD2 knockout in each artery type cannot be explained by changes in the expression of adrenergic receptors.

6) Furthermore, given they see differential effects between myocyte, where else is the SMMHC-CreERT2 expressed? Is Tamoxifen treatment equally efficient for knocking out PKD2 in all places?

We have performed additional immunofluorescence and contractility experiments. We now demonstrate that PKD2 is present in hindlimb artery myocytes of PKD2^fl/fl^ mice, but absent in hindlimb artery myocytes of PKD2^sm-/-^ mice (new Figure 1D). These data compliment those provided in the first submission which showed that PKD2 is knocked out in mesenteric artery myocytes of PKD2^sm-/-^ mice. We also now show that phenylephrine similarly constricts hindlimb arteries of PKD2^fl/fl^ and PKD2^sm-/-^ mice (new Figure 3—figure supplement 1C). These data provide additional support for our conclusion that α1 adrenergic receptors do not signal through PKD2 channels in myocytes of hindlimb arteries.

7) It has become standard to show the individual data points for each bar graph so the reader can more fully appreciate the sample size, distribution and effect sizes.

We agree and now show the individual data points on all bar graphs.

8) Although they discuss TRP channels, they should also consider the roles of Piezo1 and Piezo2. Are these expressed in myocytes? Are they dysregulated after PKD2 deletion?

There is evidence for Piezo1 expression in arterial myocytes, although Piezo1 is not required for the myogenic response in caudal and cerebral arteries^2^. We have performed additional experiments and now show that Piezo1 protein is similar in mesenteric and hindlimb arteries of PKD2^fl/fl^ and PKD2^sm-/-^ mice (new Figure 1—figure supplement 3). These data suggest that the attenuated pressure induced vasoconstriction in PKD2^sm-/-^ hindlimb arteries cannot be explained by changes in Piezo1 expression in myocytes. These new data and evidence for Piezo1 expression in arterial myocytes have now been discussed in the manuscript.

9) Similarly, GPCRs such as the angiotensin receptor and (very recently) GPR68 have been implicated in mechanosensing. Are these found in myocytes? Could PKD2 be functioning downstream? Some more discussion of what they think PKD2 is doing would be helpful.

We include new data and now show that GPR68 and angiotensin II receptor proteins are similar in mesenteric and hindlimb arteries of PKD2^fl/fl^ and PKD2^sm-/-^ mice (new Figure 1—figure supplement 3). We have expanded the Discussion section to describe evidence that these proteins are mechanosensors in smooth muscle and the possibility that PKD2 channels may be downstream effectors.

10) The summary bar graphs as provided are insufficient to evaluate the sample size, distribution and effect sizes.

As suggested, individual data points are now shown on all bar graphs.

Reviewer #2:[…] 1) Determine if a PKD2-dependent single channel conductance can be measured from the plasma membrane and compare them from the mesenteric and hindlimb smooth muscle cells.

The goals of the study were to test the hypothesis that arterial myocyte PKD2 channels regulate systemic blood pressure and to investigate mechanisms involved. The measurements you suggest would aim to determine whether the single channel conductances of swelling-activated channels in hindlimb artery myocytes and phenylephrine-activated channels in mesenteric artery myocytes are similar or different. To be certain that single channels which are measured are PKD2, we would need to show that these specific single channel conductances are absent in PKD2^sm-/-^ myocytes from each artery type. To attempt to directly address your question, we performed patch-pressure experiments in hindlimb artery myocytes using similar approaches to those we used to measure single TMEM16A channels in cerebral artery myocytes^1^. We found that negative pressure activated single non-selective channels in PKD2^fl/fl^ myocytes, although this only occurred in a small percentage (in 3 out of 30 = 10% ) of cell-attached patches. Given that the plasma membrane surface area of a cell-attached patch is ~1 µm^2^ and that of an arterial myocyte is ~3141 µm^2^ (assuming a 10 µm diameter cylinder that is 100 µm long), the low probability of obtaining pressure-sensitive channels within individual patches is not surprising. However, this creates technical challenges with performing the experiments you suggested. Because of the low probability of obtaining pressure-sensitive channels in patches, we would have to record from a very large number of both hindlimb artery and mesenteric artery myocytes (~100 of each type) to obtain sufficient recordings to characterize channel properties. We would also need to record from an equally large number of hindlimb artery and mesenteric artery myocytes of PKD2^sm-/-^ myocytes to determine whether the same single channel conductances are absent. Other challenges exist. PKD2 knockout only partially attenuated phenylephrine-induced vasoconstriction, indicating that PKD2 channel activation is only one mechanism activated by this vasoconstrictor. Phenylephrine is likely to activate several different ion channel types in myocytes through G_q11_ and via the release of intracellular Ca^2+^, which will stimulate Ca^2+^-activated channels. Phenylephrine-induced simultaneous activation of multiple different ion channel types that have different conductances will make it difficult to identify individual openings of PKD2 channels. We consider this extensive identification of single channel conductances in hindlimb and mesenteric hindlimb artery myocytes of PKD2^fl/fl^ and PKD2^sm-/-^ myocytes from two different artery types that are activated by distinct stimuli to be well beyond the scope of the current study. We also believe that this information does not alter the principal conclusions of our paper that arterial myocyte PKD2 channels increase systemic blood pressure by activating Na^+^ currents that induce membrane depolarization and vasoconstriction. Considering the large amount of work that would be involved, we consider that a future, focused study should compare the single channel properties of PKD2 channels in myocytes of different arterial beds.

2) Determine if the putative PKD2 currents are cation-selective from the mesenteric and hindlimb smooth muscle. The authors used symmetric NaCl or NaAsp solutions and conclude that this channel is Na-permeable or is "cation-selective" without demonstrating this, as either anion can permeate chloride channels in smooth muscle (PMID: 10087337). This is particularly important for the osmo-regulation experiments, as the 300-250mOs condition invariably causes an increasing in the ohmic leak currents and likely activates resident swell-activated Cl^-^ channels.

As suggested, we have performed new experiments. We now show using cation substitution that cell swelling activates a Na^+^ current in hindlimb artery myocytes and phenylephrine stimulates a Na^+^ current in mesenteric artery myocytes (new Figure 4 and Figure 6). Current activation by these stimuli is absent in the same arterial myocytes of PKD2^sm-/-^ mice. These data provide further support that swelling activates PKD2 channels in hindlimb artery myocytes and phenylephrine stimulates PKD2 channels in mesenteric artery myocytes, generating Na^+^ currents. The inclusion of these new data made previous Figure 4F and 4G redundant, so they were removed.

3) The reversal potentials and voltage-dependence are variable between the electrophysiology recordings that report the putative PKD2 channel activity. Thus, uniform steady state analysis of the current should be per An example of this is Figure 4 —figure supplement 1and Figure 6—figure supplement 1. In Figure 6—figure supplement 1the reversal potential shifts from approximately -10 mV to +10 mV between the knockout and control currents (respectively), but it is zero millivolts experiments under the same conditions. Perhaps there are differences in the relative ion permeability of these cell membranes, that is variable from cell to cell, but cannot be considered a pure PKD2 current.

Please refer to our response to your previous comment (#2) and see our new data where have performed Na^+^ substitution experiments and report current reversal potentials (new Figure 4 and Figure 6). Mean reversal potentials in mesenteric and hindlimb artery myocytes of PKD2^fl/fl^ and PKD2^sm-/-^ mice are not different. Collectively, data show that PKD2 channel activation generates Na^+^ currents in arterial myocytes that induce membrane depolarization, vasoconstriction and an increase in systemic blood pressure.

4) Whether hypertension in ADPKD is a primary vasculopathy secondary to mutations in the polycystin genes or perhaps, secondary to activation of the renin-angiotensin-aldosterone system by cyst expansion and intrarenal ischemia is unclear and thus is an intriguing question. However, it is important to point out that hypertension, not hypotension (as observed in their mouse) is associated with ADPKD in man. Thus, while the findings are descriptive of PKD2 regulation of vascular tone, it does little to enhance our understanding the disease complications associated with this channel.

We have expanded the paragraph in the Discussion section to consider these points in more detail.

The goal of our study was to investigate blood pressure regulation by arterial myocyte PKD2 channels. One finding of our paper is that inducible, smooth muscle cell-specific PKD2 channel knockout reduces systemic blood pressure. Regarding disease complications associated with this channel, we show that PKD2 is upregulated in hypertension and that smooth muscle-specific knockout causes vasodilation and reduces arterial wall remodeling and hypertension. We respectfully disagree that or study does little to enhance our understanding of disease complications associated with this channel. It was not our objective to study ADPKD or pathological effects of ADPKD-associated PKD2 channel mutations in arterial myocytes. Our demonstration that myocyte PKD2 channels regulate blood pressure is a step forward to better understanding the significance of this myocyte ion channel in cardiovascular physiology and disease.

As we discussed in the paper and as you point out, a constitutive and global PKD2 channel mutation is associated with hypertension in ADPKD patients. Here, we studied the effects of knocking out arterial myocyte PKD2 over an approximately three week time period in fully developed adult mice. There are many reasons for why an acute, myocyte-specific PKD2 knockout model and a constitutive and global PKD2 mutation have different effects on blood pressure. We refer you to our expanded Discussion section, where we have outlined these considerations in more detail. Briefly, these include differences that may occur when studying PKD2 mutations versus PKD2 knockout, global versus myocyte-specific effects, and the result of a constitutive mutation that is present since fertilization versus those of an acute (<3 week) knockout in fully developed adults. Future studies should be designed to investigate the effects of PKD2 mutations that occur in ADPKD patients on vascular myocyte function and systemic blood pressure.Reviewer #3:[…] 1) Introduction: The motivation and rationale for the study are not well-developed. For example, in the first paragraph, it is stated "However the molecular identity of arterial myocyte ion channels that regulate blood pressure, and their mechanisms of modulation in vivo (are) poorly understood." This is not entirely true. L-type voltage-dependent calcium channels in vascular smooth muscle regulate blood pressure and are the targets of a major class of anti-hypertensive drugs. Furthermore, the "in vivo" modulation of any type of ion channel is not completely understood. The current study only sheds light on PKD2 modulation in vitro, and not in vivo.

We have modified this paragraph to better outline the rationale for the study.

2) Introduction and elsewhere: It is stated a number of times that the PKD2 knockout is in "systemic artery myocytes". This not true; it is all smooth muscle (vascular and non-vascular). This should be made clear. Do the mice have any other defects?

As suggested, these statements have been modified to improve clarity. The mice do not have any obvious defects other than those described in the paper. Locomotion, cardiac function, plasma angiotensin II, aldosterone and ANP and plasma and urine electrolytes were all similar in PKD2^fl/fl^ and PKD2^sm-/-^ mice.

3) Subsection “Swelling activates PKD2 channels in hindlimb artery myocytes”: Cell-swelling-based electrophysiology data are incomplete. The authors show that cell swelling stimulates a TRP-like current in hindlimb artery myocytes, consistent with the demonstration that PKD2 plays a role in a mechanosensitive response (pressure-induced constriction) in hindlimb arteries. Although they report that PKD2 is not involved in pressure-induced constriction in mesenteric arteries, whether swelling-induced TRP-like currents are absent or present in myocytes from these arteries is not tested.

These data were provided in the first submission of the manuscript as supplemental figure 8. Due to figure rearrangement in the revised manuscript, these data are now shown as Figure 6—figure supplement 1. These results indicate that cell swelling activates a similar amplitude I_Cat_ in mesenteric artery myocytes of PKD2^fl/fl^ and PKD2^sm-/-^ mice.

4) It is unclear why isotonic Na^+^ was used. A physiological solution could have been used, so that swelling-activated (Na+) current over physiological membrane potentials can be examined. Furthermore, a similar protocol was used by Welsh et al., 2002, with the conclusion that the cation current was through TRPC6 channels. This study should be discussed.

The goal of the study was to investigate the regulation of blood pressure by PKD2 channels, which are non-selective cation channels. Isotonic Na^+^ was used to isolate currents through TRP channels and to prevent currents through K^+^ channels. We have performed additional patch-clamp experiments to measure the permeability of the swelling-activated current. These new data show that hypoosmolar solution activates a Na^+^ current (new Figure 4). As suggested, we have discussed the Welsh study, which demonstrated that myocyte TRPC6 channels contribute to the myogenic response in cerebral arteries, a vascular bed that does not regulate systemic blood pressure.

5) Results section: PE-based electrophysiology data are incomplete. The authors show that PE stimulates a TRP-like current in mesenteric artery myocytes, consistent with the demonstration that PE induces constriction of mesenteric arteries. The companion patch-clamp experiments (PE on hindlimb artery myocytes) are nowhere to be found. Since other TRP channels are present in these cells (at minimum, TRPM4, TRPC6 and TRPV4, according to their own presentation) and should be detectable under these conditions, this result is puzzling.

These data are now shown as Figure 4—figure supplement 1. They show that phenylephrine activates a similar amplitude I_Cat_ in hindlimb artery myocytes of PKD2^fl/fl^ and PKD2^sm-/-^ mice. We have also performed additional new experiments and show that phenylephrine similarly constricts hindlimb arteries of PKD2^fl/fl^ and PKD2^sm-/-^ mice (new Figure 3—figure supplement 1C). These data provide additional support for our conclusion that α1-adrenergic receptors do not signal to PKD2 channels in myocytes of hindlimb arteries.

6) Not at all clear why PKD2 KO should have a dramatic effect on myogenic tone in hindlimb (and cerebral arteries) arteries, but no effect at all in mesenteric arteries. Seemingly inexplicable results are often the takeoff point for interesting scientific stories, but they are intellectually unsatisfying if left hanging, as they are here. Again, at first blush, it makes no sense that PKD2 KO would have a dramatic effect on PE effects in mesenteric arteries, but no effect in hindlimb arteries.

We have expanded the text and provide multiple explanations why unique stimuli may activate PKD2 channels in myocytes of different arteries. Arteries within different organs are exposed to unique environments, including diverse intravascular pressures and the types and concentrations of a wide range of vasoconstrictor and vasodilator stimuli. Arteries of different organs are not the same. Clear examples include cerebral, pulmonary and coronary arteries. It would be surprising if the molecular components of signaling pathways were identical in myocytes of different arteries. The goal of our study was to investigate blood pressure regulation by arterial myocyte PKD2 channels. We found using a knockout model that the same ion channel, PKD2, is activated by distinct vasoconstrictor stimuli in two different vascular beds. It was not a goal of our study to identify the differential signaling mechanisms involved in PKD2 activation in these different vascular beds. As you mention, we consider this finding to be a “takeoff point” for future studies and well beyond the scope of the current manuscript.

References

1. Bulley S, Neeb ZP, Burris SK, Bannister JP, Thomas-Gatewood CM, Jangsangthong W and Jaggar JH. TMEM16A/ANO1 channels contribute to the myogenic response in cerebral arteries. Circ Res. 2012;111:1027-1036.

2. Retailleau K, Duprat F, Arhatte M, Ranade SS, Peyronnet R, Martins JR, Jodar M, Moro C, Offermanns S, Feng Y, Demolombe S, Patel A and Honore E. Piezo1 in Smooth Muscle Cells Is Involved in Hypertension-Dependent Arterial Remodeling. Cell Rep. 2015;13:10

[Editors' note: the author responses to the re-review follow.]

Essential revisions:1) The representative traces in Figure 4C can be confusing when compared to the Gd-sensitive currents plotted in Figure 4D. Please show traces for Gd for both conditions studied in Figure 4C and explicitly point out in the text what is shown in Figure 4C and D. Also, the graphs in Figures 4D and Figure 4—figure supplement 1B are confusing because only data for -100 and +100 are shown, yet they appear on a continuous x-axis labeled voltage. The x axis should be removed, and data plotted at bar graphs with conditions clearly labeled.

In Figure 4, we have included a new panel illustrating representative traces for Gd^3+^-sensitive currents in hypotonic solution in *Pkd2^fl/fl^* and *Pkd2* smKO myocytes (Figure 4D). We have referred to this figure in the Results section and the figure legend. In Figure 4E (previously 4D), the x-axis has been removed from Figure 4E and the y-axis moved to the left for better visibility. The y-axis in Figure 4—figure supplement 1B has also been moved to the left of the graph.

2) It appears that some of the changes made in revision were done hastily, as there are ambiguous statements and grammatical errors in the red text. For example, these lines from the methods are non-sequiturs "Equimolar N-methyl-glucamine (NMDG) was substituted for Na^+^. Liquid junction potential was measured experimentally." Substituted for which solution? What was the LJ potential then if it was measured and what was done with that measurement? Please take your time revising the text and figures to make sure everything will be clear to readers.

We have made sure there are no grammatical errors or ambiguous statements in the text. In the Materials and methods section and Results section, we have described how liquid junction potentials were measured and used to correct reversal potential values recorded in patch-clamp experiments. The results for the liquid junction potential measurements are now in the Results section.

[Editors' note: further revisions were requested prior to acceptance, as described below.]

Your revised manuscript has addressed our specific requests of the reviewing editor and reviewers. However, in evaluating the full manuscript it became evident that the conceptual problem you have nicely corrected in Figure 4E of the revised manuscript needs to also be applied to Figure 4—figure supplement 1, Figure 6D and Figure 6—figure supplement 1. We request that you make the appropriate changes to those remaining figures before publication.

As requested, we have modified the format of the graphs shown in Figure 4—figure supplement 1, Figure 6D and Figure 6—figure supplement 1 to match that of the graph shown in Figure 4E. These new figures have been uploaded.